METHODS AND RESOURCES

# Expansion of RiPP biosynthetic space through integration of pan-genomics and machine learning uncovers a novel class of lanthipeptides

**Alexander M. Kloosterman**[1], **Peter Cimermancic**[2☉], **Somayah S. Elsayed**[1☉], **Chao Du**[1], **Michalis Hadjithomas**[3¤], **Mohamed S. Donia**[4], **Michael A. Fischbach**[5], **Gilles P. van Wezel**[1,6]*, **Marnix H. Medema**[7]*

**1** Institute of Biology, Leiden University, the Netherlands, **2** Verily Life Sciences, South San Francisco, CA, United States of America, **3** DOE Joint Genome Institute, Walnut Creek, CA, United States of America, **4** Department of Molecular Biology, Princeton University, NJ, United States of America, **5** Department of Bioengineering, Stanford University, CA, United States of America, **6** Netherlands Institute for Ecology (NIOO-KNAW), Wageningen, the Netherlands, **7** Bioinformatics group, Wageningen University, the Netherlands

☉ These authors contributed equally to this work.
¤ Current address: LifeMine Therapeutics, Cambridge, Massachusetts, United States of America
* g.wezel@biology.leidenuniv.nl (GPvW); marnix.medema@wur.nl (MHM)

**Data Availability Statement:** The source code of decRiPPter is freely available online at https://github.com/Alexamk/decRiPPter. Results of the

## Abstract

Microbial natural products constitute a wide variety of chemical compounds, many which can have antibiotic, antiviral, or anticancer properties that make them interesting for clinical purposes. Natural product classes include polyketides (PKs), nonribosomal peptides (NRPs), and ribosomally synthesized and post-translationally modified peptides (RiPPs). While variants of biosynthetic gene clusters (BGCs) for known classes of natural products are easy to identify in genome sequences, BGCs for new compound classes escape attention. In particular, evidence is accumulating that for RiPPs, subclasses known thus far may only represent the tip of an iceberg. Here, we present decRiPPter (Data-driven Exploratory Class-independent RiPP TrackER), a RiPP genome mining algorithm aimed at the discovery of novel RiPP classes. DecRiPPter combines a Support Vector Machine (SVM) that identifies candidate RiPP precursors with pan-genomic analyses to identify which of these are encoded within operon-like structures that are part of the accessory genome of a genus. Subsequently, it prioritizes such regions based on the presence of new enzymology and based on patterns of gene cluster and precursor peptide conservation across species. We then applied decRiPPter to mine 1,295 *Streptomyces* genomes, which led to the identification of 42 new candidate RiPP families that could not be found by existing programs. One of these was studied further and elucidated as a representative of a novel subfamily of lanthipeptides, which we designate class V. The 2D structure of the new RiPP, which we name pristinin A3 (**1**), was solved using nuclear magnetic resonance (NMR), tandem mass spectrometry (MS/MS) data, and chemical labeling. Two previously unidentified modifying enzymes are proposed to create the hallmark lanthionine bridges. Taken together, our work highlights how novel natural product families can be discovered by methods going beyond sequence similarity searches to integrate multiple pathway discovery criteria.

data analysis are available online at https://decrippter.bioinformatics.nl. All training data and code used to generate these, as well as outputs of the data analyses, are available on Zenodo at doi:10.5281/zenodo.3834818. NMR data will be made available on http://www.np-mrd.org/ when this database opens up for submissions.

**Funding:** The work of AK was funded by a grant to GPvW from the Netherlands Organization for Scientific Research (NWO), project nr 731.014.206. The funders had no role in study design, data collection and analysis, decision to publish, or preparation of the manuscript.

**Competing interests:** I have read the journal's policy and the authors of this manuscript have the following competing interests: P.C. is currently an employee of Verily Life Sciences. M.H. is currently an employee of LifeMine Therapeutics. M.S.D. is a member of the Scientific Advisory Board of DeepBiome Therapeutics. M.A.F. is a cofounder and director of Federation Bio. M.H.M. is on the scientific advisory board of Hexagon Bio and co-founder of Design Pharmaceuticals.

**Abbreviations:** BBH, bidirectional best hit; BGC, biosynthetic gene cluster; BUSCO, Benchmarking set of Universal Single-Copy Orthologs; CDS, calibrant delivery system; CID, collision induced dissociation; COG, cluster of orthologous genes; CORASON, CORe Analysis of Syntenic Orthologs to prioritize Natural Product-Biosynthetic Gene Clusters; COSY, correlation spectroscopy; decRiPPter, Data-driven Exploratory Class-independent RiPP TrackER; Dha, dehydroalanine; Dhb, dehydrobutyrine; DMSO-$d_6$, deuterated dimethylsulfoxide; ESI, electrospray ionization; HMBC, heteronuclear multiple bond correlation; HPLC-MS, high-performance liquid chromatography-mass-spectrometry; HSQC, heteronuclear single quantum coherence; IAA, iodoacetamide; Lan, lanthionine; LC-MS, liquid chromatography–mass spectrometry; MCL, Markov Clustering Algorithm; MIBiG, Minimum Information about a Biosynthetic Gene cluster; MS/MS, tandem mass spectrometry; NMe2-MeLan, $N$,$N$-dimethyl-β-methyllanthionine; NMMP, NH4-based Minimal Medium with Phosphate; NOESY, nuclear overhauser effect spectroscopy; NRP, nonribosomal peptide; NRPS, nonribosomal peptide synthetase; OEP, omega-ester containing peptide; ORF, open reading frame; PDA, photodiode array detector; PK, polyketides; RiPP, ribosomally synthesized and post-translationally modified peptide; ROC, receiver operating characteristic; SAM, $S$-adenosylmethionine; SFM, mannitol soya flour; SVM, Support Vector

## Introduction

The introduction of antibiotics in the 20th century contributed hugely to extend the human life span. However, the increase in antibiotic resistance and the concomitant steep decline in the number of new compounds discovered via high-throughput screening [1,2] means that we again face huge challenges to treat infections by multidrug resistant bacteria [3]. The low return of investment of high-throughput screening is due to dereplication, in other words, the rediscovery of bioactive compounds that have been identified before [4,5]. A revolution in our understanding was brought about by the development of next-generation sequencing technologies. Actinobacteria are the most prolific producers of bioactive compounds, including some two-thirds of the clinical antibiotics [6,7]. Mining of the genome sequences of these bacteria revealed a huge repository of previously unseen biosynthetic gene clusters (BGCs), highlighting that their potential as producers of bioactive molecules had been grossly underestimated [6,8,9]. However, these BGCs are often not expressed under laboratory conditions, most likely because the environmental cues that activate their expression in their original habitat are missing [10,11]. To circumvent these issues, a common strategy is to select a candidate BGC and force its expression by expression of the pathway-specific activator or via expression of the BGC in a heterologous host [12]. However, these methods are time-consuming, while it is hard to predict the novelty and utility of the compounds they produce.

To improve the success of genome mining-based drug discovery, many bioinformatic tools have been developed for identification and prioritization of BGCs. These tools often rely on conserved genetic markers present in BGCs of certain natural products, such as polyketides (PKs), non-ribosomal peptides (NRPs), and terpenes [13–15]. While these methods have unearthed vast amounts of uncharacterized BGCs, they further expand on previously characterized classes of natural products. This raises the question of whether entirely novel classes of natural products could still be discovered. A few genome mining methods, such as ClusterFinder [16] and EvoMining [17,18], have tried to tackle this problem. These methods either use criteria true of all BGCs or build around the evolutionary properties of gene families found in BGCs, rather than using BGC-class-specific genetic markers. While the lack of clear genetic markers may result in a higher number of false positives, these methods have indeed charted previously uncovered biochemical space and led to the discovery of new natural products.

One class of natural products whose expansion has been fueled by the increased amount of genomic sequences available is that of the ribosomally synthesized and post-translationally modified peptides (RiPPs) [19]. RiPPs are characterized by a unifying biosynthetic theme: A small gene encodes a short precursor peptide, which is extensively modified by a series of enzymes that typically recognize the N-terminal part of the precursor called the leader peptide, and finally cleaved to yield the mature product [20]. Despite this common biosynthetic logic, RiPP modifications are highly diverse. The latest comprehensive review categorizes RiPPs into roughly 20 different classes [19], such as lanthipeptides, lasso peptides, and thiopeptides. Each of these classes is characterized by one or more specific modifications, such as the thioether bridge in lanthipeptides or the knot-like structure of lasso peptides. Despite the extensive list of known classes and modifications, new RiPP classes are still being found. Newly identified RiPP classes often carry unusual modifications, such as D-amino acids [21], addition of unnatural amino acids [22,23], β-amino acids [24], or new variants of thioether crosslinks [25,26]. These discoveries strongly indicate that the RiPP genomic landscape remains far from completely charted and that novel types of RiPPs with new and unique biological activities may yet be uncovered. However, RiPPs pose a unique and major challenge to genome-based pathway identification attempts: Unlike in the case of nonribosomal peptide synthetases (NRPSs) and polyketide synthetases (PKSs), there are no universally conserved enzyme

Machine; TOCSY, total correlation spectroscopy; trueCOG, true cluster of orthologous genes.

families or enzymatic domains that are found across all RiPP pathways. Rather, each class of RiPPs comprises its own unique set of enzyme families to post-translationally modify the precursor peptides belonging to that class. Hence, while BGCs for known RiPP classes can be identified using conventional genome mining algorithms, a much more elaborate strategy is required to automate the identification of novel RiPP classes.

Several methods have made progress in tackling this challenge. "Bait-based" approaches such as RODEO [26–31] and RiPPer [32] identify RiPP BGCs by looking for homologs of RiPP modifying enzymes of interest and facilitate identifying the genes encoding these enzymes in novel contexts to find many new RiPP BGCs. A study was also described using a transporter gene as a query that is less dependent on a specific RiPP subclass [33]. However, these methods still require a known query gene from a known RiPP subclass. Another tool recently described, NeuRiPP, is capable of predicting precursors independent of RiPP subclass but is limited to precursor analysis [34]. Yet another tool, DeepRiPP, can detect novel RiPP BGCs that are chemically far removed from known examples but is mainly designed to identify new members of known classes [35]. In the end, an algorithm for the discovery of BGCs encoding novel RiPP classes will need to integrate various sources of information to reliably identify genomic regions that are likely to encode RiPP precursors along with previously undiscovered modifying enzymes.

Here, we present decRiPPter (Data-driven Exploratory Class-independent RiPP TrackER), an integrative algorithm for the discovery of novel classes of RiPPs, without requiring prior knowledge of their specific modifications or core enzymatic machinery. DecRiPPter employs a Support Vector Machine (SVM) classifier that predicts RiPP precursors regardless of RiPP subclass, and combines this with pan-genomic analysis to identify which putative precursor genes are located within specialized genomic regions that encode multiple enzymes and are part of the accessory genome of a genus. Sequence similarity networking of the resulting precursors and gene clusters then facilitates further prioritization. Applying this method to the gifted natural product producer genus *Streptomyces*, we identified 42 new RiPP family candidates. Experimental characterization of a widely distributed candidate RiPP BGC led to the discovery of a novel lanthipeptide that was produced by a previously unknown enzymatic machinery.

## Results

### RiPP BGC discovery by detection of genomic islands with characteristics typical of RiPP BGCs

Given the promise of RiPPs as a source for novel natural products, we set out to construct a platform to facilitate identification of novel RiPP classes. Since no criteria could be used that are specific for individual RiPP classes, we used 3 criteria that generally apply to RiPP BGCs: (1) they contain one or more open reading frames (ORFs) for a precursor peptide; (2) they contain genes encoding modifying machinery in an operon-like gene cluster together with precursor gene(s); and (3) they have a sparse distribution within the wider taxonomic group in which they are found. To focus on novel RiPP classes, we added a fourth criterion: (4) they have no direct similarity to BGCs of known classes (Fig 1).

For the first criterion, we trained an SVM-based classifier to distinguish between RiPP precursors and other peptides. A collection of 175 known RiPP precursors, gathered from RiPP clusters from the Minimum Information about a Biosynthetic Gene cluster (MIBiG) repository [36,37] was used as a positive training set (S1A Data). For the negative training set, we generated a set of 20,000 short non-precursor sequences, consisting of 10,000 randomly selected short proteins (<175 amino acids long) from Uniprot without measurable similarity to RiPP precursors (representative of gene encoding proteins but not RiPP precursors), and

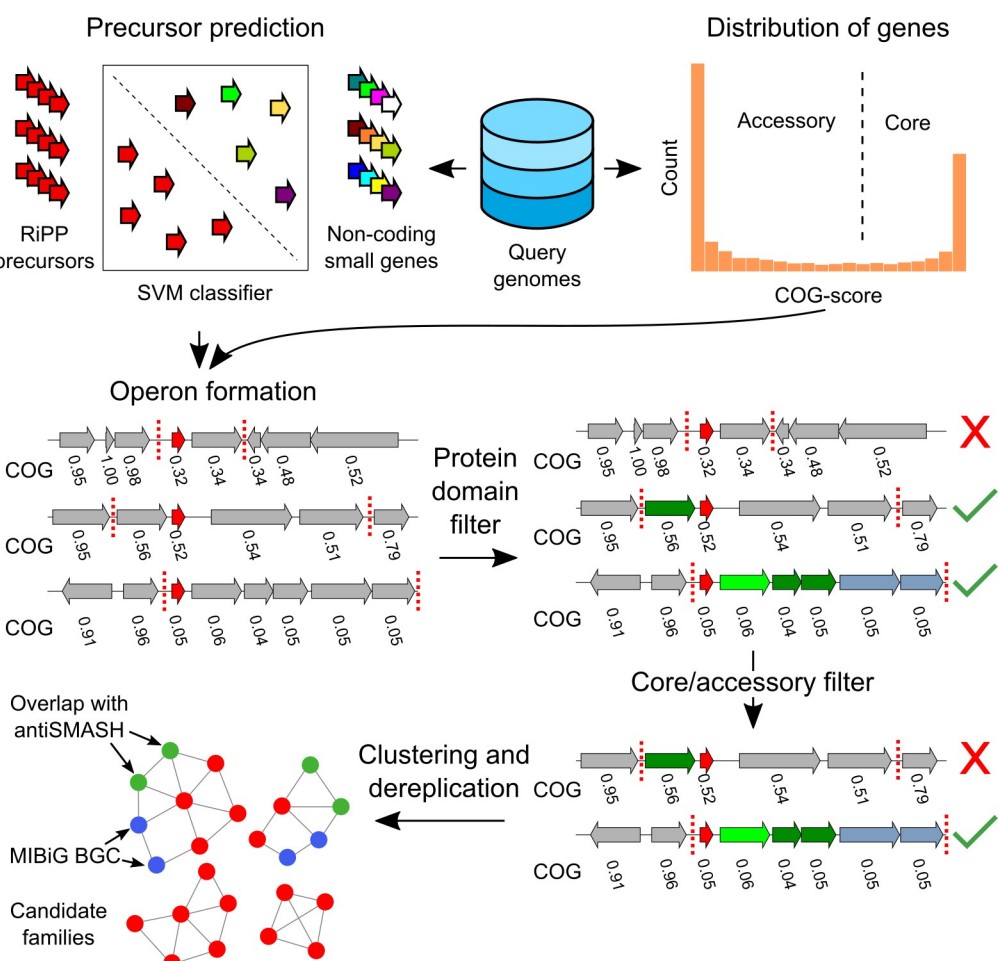

**Fig 1. decRiPPter pipeline for the detection of novel RiPP families.** The SVM classifiers is used to identify all candidate RiPP precursors in a given group of genomes, using all predicted proteins smaller than 100 amino acids. The gene clusters formed around the precursors are analyzed for specific protein domains. In addition, all COG scores are calculated to act as an additional filter and to aid in gene cluster detection. The remaining gene clusters are clustered together and with MIBiG gene clusters to dereplicate and organize the results. In addition, overlap with antiSMASH detected BGCs is analyzed. BGC, biosynthetic gene cluster; COG, cluster of orthologous genes; decRiPPter, Data-driven Exploratory Class-independent RiPP TrackER; MIBiG, Minimum Information about a Biosynthetic Gene cluster; RiPP, ribosomally synthesized and post-translationally modified peptide; SVM, Support Vector Machine.

10,000 translated intergenic sequences between a stop codon and the next start codon of sizes 30 to 300 nt taken from 10 genomes across the bacterial tree of life (representative of spurious ORFs that do not encode proteins). From both positive and negative training set sequences, 36 different features were extracted describing the amino acid composition and physicochemical properties of the protein/peptide sequences, as well as localized enrichment of amino acids prone to modification by modifying enzymes. Based on these, several SVMs were trained (Materials and methods), of which the average prediction score was used to classify peptides. To make sure that this classifier could predict precursors independent of RiPP subclass, we trained it on all possible subsets of the positive training set in which one of the RiPP subclasses was entirely left out (a strategy we termed leave-one-class-out cross-validation). Typically, the classifier was still capable of predicting the class that was left out, with an area-under-receiver operating characteristics curve of 0.955. To validate the classifier, we used it to classify precursor hits from the various RiPP mining studies performed using RODEO [26–31]. In general,

66.7% of all precursors identified by RODEO's SVMs were scored as positive by decRiPPter's SVMs (S1B Data). This shows that, for known RiPP classes, the classifier described here is well capable of detecting the majority of precursor peptides, although it is, unsurprisingly, outperformed by the dedicated, class-specific SVMs of RODEO.

For the second criterion, we made use of the fact that the majority of RiPP BGCs appear to contain the genes encoding the precursor and the core biosynthetic enzymes in the same strand orientation within close intergenic distance (81.6% of MIBiG RiPPs). Therefore, candidate gene clusters are formed from the genes that appear to reside in an operon with predicted precursor genes, based on intergenic distance and the cluster of orthologous genes (COG) scores calculated (see description below, Materials and methods, S1 and S2 Figs). These gene clusters were then analyzed for protein domains that could constitute the modifying machinery (Fig 1B). Rather than restricting ourselves to specific protein domains, we constructed a broad dataset of Pfam and TIGRFAM domains that are linked to an E.C. number using InterPro mappings [38]. This dataset was extended with a previously curated set of Pfam domains found to be prevalent in the positive training set of the ClusterFinder algorithm [16], and manually curated, resulting in a set of 4,131 protein domains. We also constructed Pfam [39] and TIGRFAM [40] domain datasets of transporters, regulators, and peptidases, as well as a dataset consisting of known RiPP modifying domains to provide more detailed annotation and allow specific filtering of RiPP BGCs based on the presence of each of these types of Pfam domains (S2 Data).

For the third criterion, we sought to distinguish specialized genomic regions from conserved genomic regions. Indeed, most BGCs are sparingly distributed among genomes, with even closely related strains showing differences in their BGC repertoires [41–43]. We therefore developed an algorithm that separates the "core" genome from the "accessory" genome, by comparing all genes in a group of query genomes from the same taxon (typically a genus), and identifying the frequency of occurrence of each gene within that group of genomes (Fig 1C and S2 Fig). For the purpose of comparing genes between genomes, we reasoned that it was more straightforward to identify groups of functionally closely related genes that also include recent paralogues, due to the complexities of dealing with orthology relationships across large numbers of genomes (especially for biosynthetic genes that are known to have a discontinuous taxonomic distribution and may undergo frequent duplications [44]). Therefore, decRiPPter first identifies the distribution of sequence identity values of protein-coding genes that can confidently be assigned to be orthologs and uses this distribution to find groups of genes across genomes with ortholog-like mutual similarity. First, a set of high-confidence orthologs, called true conserved orthologous genes (trueCOGs) are identified based on 2 criteria: (1) they should be bidirectional best hits (BBH) between all genome pairs; and (2) their 2 flanking genes should also be BBHs between all genome pairs [45]. In other words, decRiPPter looks for sets of 3 contiguous genes that are highly conserved in both sequence identity and synteny among all analyzed genomes, using DIAMOND [46]. The center genes of these gene triplets are themselves conserved, and have conserved surrounding genes, making it highly likely that they are orthologous to one another. These center genes were therefore considered trueCOGs. While this list of trueCOGs contains high-confidence orthologs, the criteria for orthology set here are strict, and many orthologs are missed by only considering orthologs based on BBHs [47]. We therefore further expanded the list of homologs with ortholog-like similarity by dynamically determining a cutoff between each genome pair based on the similarity of the trueCOGs shared between those genomes. This cutoff is used to find all highly similar gene pairs. Considering that only sequence identity is used as a cutoff here, these gene pairs are either orthologs or paralogs. The identified gene pairs are then clustered with the Markov Clustering Algorithm (MCL [48,49]) into COGs. The number of COG members found for

**Table 1. Correlation between the strictness of the filter used on the identified gene clusters and the saturation of RiPP BGCs.**

| Filter | Filter details | Number of detected gene clusters | Number of detected gene clusters overlapping antiSMASH RiPP BGCs | Percentage of detected gene clusters overlapping RiPP BGCs |
|---|---|---|---|---|
| None | - | 718,268 | 5,908 | 0.8% |
| Mild | Gene cluster COG score: $<= 0.25$<br>In the gene cluster:<br>• $>= 3$ genes<br>• $>= 2$ biosynthetic genes<br>In or around the gene cluster:<br>• $>= 1$ transporter gene | 21,419 | 1,678 | 7.8% |
| Strict | Gene cluster COG score: $<= 0.10$<br>In the gene cluster:<br>• $>= 3$ genes<br>• $>= 2$ biosynthetic genes<br>In or around the gene cluster:<br>• $>= 1$ transporter gene<br>• $>= 1$ regulatory gene<br>• $>= 1$ peptidase gene | 2,471 | 357 | 14.4% |

BGC, biosynthetic gene cluster; COG, cluster of orthologous genes; RiPP, ribosomally synthesized and post-translationally modified peptide.

each gene is divided by the number of genomes in the query to get a COG score ranging from 0 to 1, reflecting how widespread the gene is across the set of query genomes (Materials and methods, S2 Fig). To validate our calculations, we analyzed the COG-scores of the highly conserved single-copy BUSCO (Benchmarking set of Universal Single-Copy Orthologs) gene set from OrthoDB [50–52], as well as the COG-scores of the genes in the gene clusters predicted by antiSMASH. In line with our expectations, homologs of the BUSCO gene set averaged COG-scores of 0.95 (S3D Fig), while the COG-scores of the antiSMASH gene clusters were much lower, averaging $0.311 \pm 0.249$ for all BGCs, and $0.234 \pm 0.166$ for RiPP BGCs (S3C Fig). While the COG-scoring method requires a group of genomes to be analyzed rather than a single genome, we believe that the extra calculation significantly contributes in filtering false positives (Table 1, S3 Fig). In addition, the COG scores aid in the gene cluster identification based on the assumption that gene clusters are generally sets of genes with similar absence/presence patterns across species (Materials and methods).

For the final criterion, the algorithm dereplicates the identified clusters by comparing them to known RiPP BGCs. All putative BGCs are clustered based on domain content and precursor similarity using sequence similarity networking [53] and compared to known RiPP BGCs from MIBiG [36,37]. In addition, the overlap between predicted RiPP BGCs and gene clusters found by antiSMASH [13,54] is determined (Fig 1).

## decRiPPter identifies 42 candidate novel RiPP classes in *Streptomyces*

While RiPPs are found in many different microorganisms, their presence in streptomycetes reflects perhaps the most diverse array of RiPP classes within a single genus. Streptomyce*s* produce a broad spectrum of RiPPs, such as lanthipeptides [55], lasso peptides [28], linear azol (in)e-containing peptides (LAPs) [56], thiopeptides [57], thioamide-containing peptides [32], and bottromycins [58–60]. Their potential as RiPP producers is further highlighted by a recent study showcasing the diversity of lanthipeptide BGCs in *Streptomyces* and other actinobacteria [61]. Even though any genus or set of genomes can be analyzed by the decRiPPter pipeline, we

hypothesized streptomycetes to be a likely source of novel RiPP classes and sought to exhaustively mine it.

We started by running the pipeline described above on all publicly available *Streptomyces* genomes (1,295 genomes) from NCBI (S1 Table). Due to computational limits, the genomes were split into 10 randomly selected groups to calculate the frequency of distribution of each gene (COG-scores). In general, the number of genomes that could be grouped together and the resulting cutoffs were found to vary with the amount of minimum trueCOGs required (S4A Fig). To make sure that as many genomes as possible could be compared at once, we set the cutoff for minimum number of trueCOGs at 10. Despite the low cutoff, the distribution of similarity scores between genome pairs still resembled a Gaussian distribution (S4B Fig). The bimodal distribution of the resulting COG-scores showed that the majority of the genes were either conserved in only a small portion of the genomes or present in almost all genomes (S3A Fig).

We then scanned all predicted products of genes as well as predicted ORFs in intergenic regions shorter than 100 amino acids (total $7.19 * 10^7$) with the SVM-based classifier. While by far most of the queries scored below 0.5, a peak of queries scoring from 0.9 to 1.0 was observed (S3B Fig). Seeking to be inclusive at this stage, we set the cutoff at 0.9, resulting in $1.32*10^6$ candidate precursors passing this initial filter, thus filtering out 98.2% of all candidates. Eliminating candidate precursors whose genes were completely overlapping reduced the number to $8.17*10^5$ precursors (1.1%). As a comparison, all ORFs were also analyzed by NLPPrecursor and NeuRiPP (S5 Fig) [34,35], and overlapping hits were removed as was done with decRiPPter's hits. For all 3 tools, a large number of candidate precursors were hits: NLPPrecursor scored the most ($4.4*10^6$), and NeuRiPP the least ($4.3*10^5$). Surprisingly, the 3 tools showed little overlap in positive hits ($1.1*10^4$). Considering that NLPPrecursor was parametrized for the detection of precursors of known classes and NeuRiPP appeared to be more strict (while our goal was to be more exploratory), we continued with decRiPPter's hits. In principle, the precursor-peptide-finding module of decRiPPter could easily be replaced by, e.g., NeuRiPP in future analyses for which this would be desirable.

We noticed that the majority of the precursor hits of decRiPPter were not found by Prodigal but were extracted from intergenic regions ($6.6*10^5$ intergenic, $1.6*10^5$ from Prodigal). A GC-plot analysis of 112 hits of both intergenic and Prodigal-detected genes showed that only 5% to 10% of the intergenic hits showed a GC-plot with clear distinctions between the first, second, and third codon position, while the majority of Prodigal-detected genes had the same distinction (S6 Fig). These intergenic regions are likely a source of many false positives, and for a more conservative approach, one could choose to ignore intergenic hits altogether. Since our aim was to conduct an explorative study to detect novel classes, and gene-finding algorithms do frequently miss precursor genes, we chose to continue with all the precursors hits found here.

In our analyses, we found that the majority of RiPP BGCs contain the majority of biosynthetic genes on the same strand orientation as the precursor (MIBiG: 81.6%; antiSMASH RiPP BGCs: 73.1%). We therefore formed gene clusters using only the genes on the same strand as the predicted precursor. To create a training set, we divided all known RiPP BGCs and all antiSMASH RiPP BGCs found in the analyzed genome sequences into sections where each section contained only genes on the same strand. The core section was defined as the section that contained the most biosynthetic genes as detected by antiSMASH or as annotated in the MIBiG database. These sections were used as training sets to fine-tune distance and COG cutoffs for our gene cluster methods.

In a simple gene cluster method, genes were joined only using the intergenic distances as a cutoff. Using this method, we found that at a distance of 750 nucleotides, all MIBiG core

sections were covered, and 91% of all antiSMASH core sections (S7A and S7B Fig). However, using only distance may cause the gene cluster formation to overshoot into regions not associated with the BGC (e.g., S1 Fig). We therefore created an alternative method called the "island method." In this method, each gene is first joined with immediately adjacent genes that lie in the same strand orientation and have very small intergenic regions (≤50 nucleotides) to form islands. These islands may subsequently be combined if they have similar average COG-scores (Materials and methods). We found that with this method, we could confidently cover our validation set, while slightly reducing the average size of the gene clusters (3.73 ± 3.75 versus 3.44 ± 3.53; S7C-S7E Fig). In addition, the variation of the COG scores within the gene clusters decreased, suggesting that fewer housekeeping genes would be added to detected BGCs (S7F Fig).

Overlapping gene clusters were fused, resulting in $7.18 * 10^5$ gene clusters. To organize the results, all clusters were paired if their protein domain content was similar (Jaccard index of protein domains; cutoff: 0.5) and at least one of their predicted precursors showed sequence similarity (NCBI blastp; bitscore cutoff: 30). These cutoffs were used to distinguish between different RiPP subclasses (S8 Fig). Clustering these pairs with MCL created 45,727 "families" of gene clusters, containing 312,163 gene clusters, while the remaining 406,105 gene clusters were left ungrouped.

Analysis of overlap between decRiPPter clusters and BGCs predicted by antiSMASH revealed that 5,908 clusters overlapped, constituting 78% of antiSMASH hits. The majority of BGCs previously detected by RODEO were also overlapping (84%, S1C Data). Most of the antiSMASH hits missed belonged to the bacteriocin family, which do not necessarily encode a small precursor peptide (S1D Data). The remainder of missed hits are likely due to precursor genes not being on the same strand as the genes encoding the biosynthetic machinery or due to precursor genes missed by decRiPPter's SVM-based classifier. The hits overlapping with antiSMASH constituted only 0.8% of all decRiPPter clusters (Table 1, row 2). To further narrow down our results, we applied several filters to increase the saturation of RiPP BGCs in our dataset. A mild filter, limiting the average COG score to 0.25 and requiring 2 biosynthetic genes and a gene encoding a transporter, increased the fraction of overlapping RiPP BGCs to 7.8% (Table 1, row 3). When only clusters associated with genes for a predicted peptidase and a predicted regulator were considered, and the average COG score was limited to 0.1, the fraction increased further to 14.4% (Table 1, row 4). While many antiSMASH RiPP BGCs were filtered out in the process (and, by extension, many unknown RiPP BGCs were likely also filtered out this way), we felt our odds of discovering novel RiPP families were highest when focusing on the dataset with the highest fraction of RiPP BGCs, and therefore applied the strict filter. The remaining 2,471 clusters of genes were clustered as described above. Since our efforts were aimed at finding new gene cluster families, we discarded groups of clusters with fewer than 3 members, leaving 1,036 gene clusters in 187 families. Families in which more than half of the gene clusters overlapped with antiSMASH non-RiPP BGCs were discarded as well, leaving only known RiPP families and new candidate RiPP families (893 gene clusters, 151 families; Fig 2). While this step eliminated BGCs for hybrids of RiPP and non-RiPP pathways, we felt this filter was necessary to reduce the number of false positives in our dataset, especially considering the rarity of these hybrid BGCs.

Roughly a third (272) of the remaining gene clusters were members of known families of RiPPs, including lasso peptides, lanthipeptides, thiopeptides, bacteriocins, and microcins. In addition, many of the other candidate clusters (55) contained genes common to known RiPP BGCs, such as those encoding YcaO cyclodehydratases and radical S-adenosylmethionine (SAM)-utilizing proteins (Fig 2). These gene clusters were not annotated as RiPP gene clusters

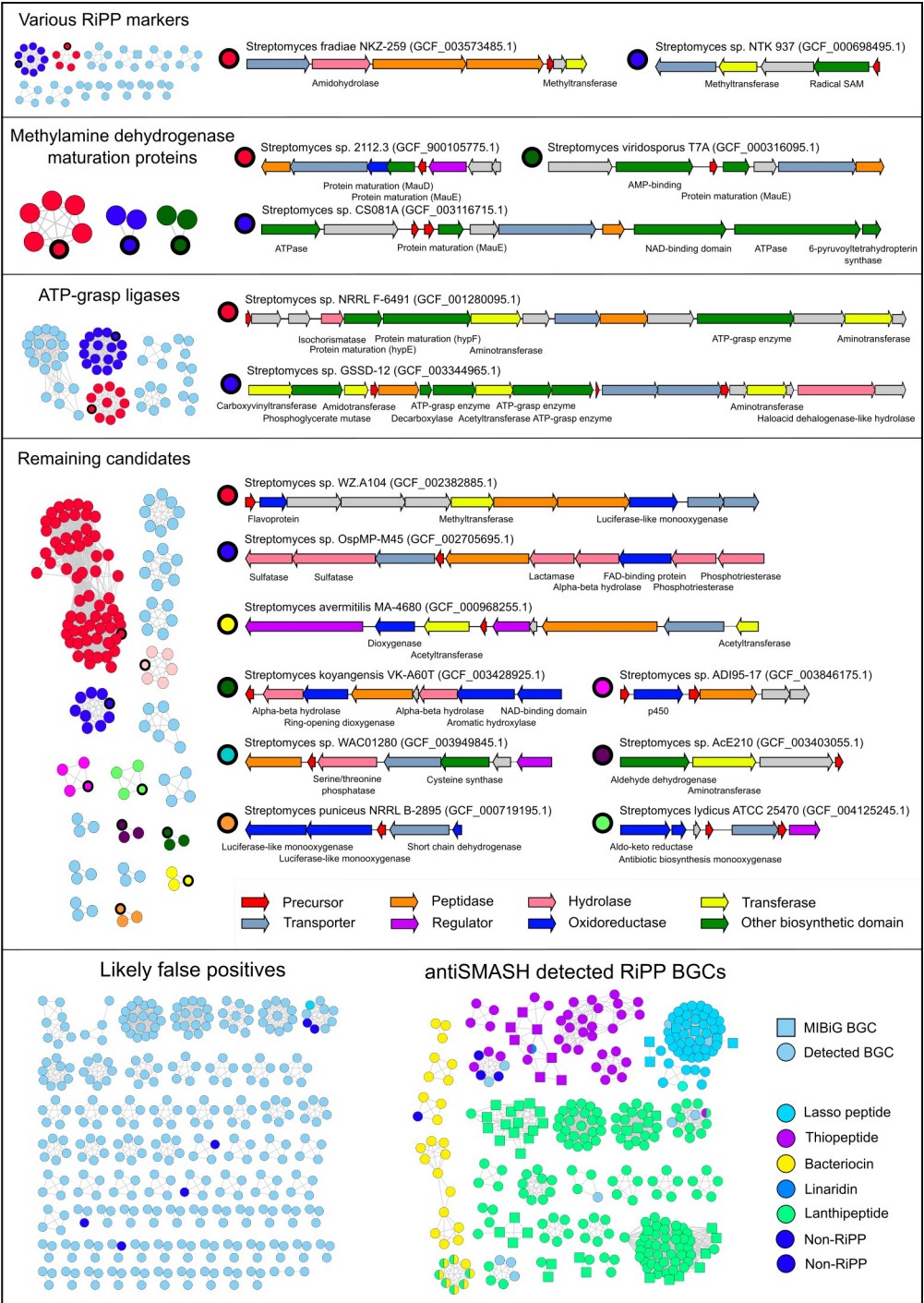

**Fig 2. decRiPPter finds 42 candidate RiPP families with a large variety of encoded modifying enzymes and precursors.** Gene clusters found in 1,295 *Streptomyces* genomes were passed through a strict filter and grouped together. Arrow colors indicate enzyme family of the product, and the description of the putative gene products is given below the arrows. Roughly a third of the remaining candidates overlapped with or were similar to RiPP BGCs predicted by antiSMASH. Another third of the remaining candidates were discarded as likely false positives. Of the remaining 42 candidate RiPP families, 15 example gene clusters are displayed. BGC, biosynthetic gene cluster; decRiPPter, Data-driven Exploratory Class-independent RiPP TrackER; RiPP, ribosomally synthesized and post-translationally modified peptide.

by antiSMASH, but the presence of these genes alone or in combination with a suitable precursor can be used as a lead to find novel RiPP gene clusters [24,32].

Each remaining family of gene clusters was manually investigated to filter out likely false positives from the candidates. A set of general guidelines followed can be found in the Materials and methods. Common reasons to discard gene clusters were functional annotations of candidate precursors as having a non-precursor function (e.g., homologous to ferredoxin or LysW [62]), annotations of the genes within a gene cluster related to primary metabolism (e.g., genes for cell-wall modifying enzymes), or other abnormalities (e.g., large intergenic gaps or very large gene cluster of more than 50 genes). Several modifying enzymes belonging to the candidate families were homologous to gene products involved in primary metabolism, such as 6-pyruvoyltetrahydropterin synthase or phosphoglycerate mutase. Given the low distribution (COG scores) of the genes encoding these enzymes, it seemed more likely to us that they were adapted from primary metabolism to play a role in secondary metabolism [17]. We therefore only discarded a gene cluster family if multiple clear relations to a known pathway were found. The remaining 42 candidate families were further grouped together into broader classes depending on whether a common enzyme was found (Fig 2).

A large group of families all contained one or more genes for ATP-grasp enzymes. ATP-grasp enzymes are all characterized by a typical ATP-grasp-fold, which binds ATP, which is hydrolyzed to catalyze a number of different reactions. As such, these enzymes have a wide variety of functions in both primary and secondary metabolism, and their genes are present in a many different genomic contexts [63]. Involvement of ATP-grasp enzymes in RiPP biosynthesis has been reported for microviridin [64] and other omega-ester containing peptides (OEPs) [65], and for pheganomycin [22], where they catalyze macrocyclization and peptide ligation, respectively. The ATP-grasp enzymes involved in the biosynthesis of these products did not show direct similarity to any of the ATP-grasp ligases of these candidates, however, suggesting that these belong to yet to be uncovered biosynthetic pathways.

Among the candidate families were 3 families that contained homologs to *mauE*, and one that additionally contained a homolog of *mauD*. The proteins encoded by these genes, along with other proteins encoded in the *mau* gene cluster, are known to be involved in the maturation of methylamine dehydrogenase, which is required for methylamine metabolism. MauE in particular has been speculated to play a role in the formation of disulfide bridges in the β-subunit of the protein, while the exact function of MauD remains unclear [66]. As no other orthologs of the *mau* cluster were found within the genomes of *Streptomyces* sp. 2112.3, *Streptomyces viridosporus* T7A, or *Streptomyces* sp. CS081A, it is unlikely that these proteins carry out this function. Rather, the presence of these genes in a putative RiPP BGC suggests that they play a role in modification of RiPP precursors. Supporting this hypothesis, each of these gene clusters contained a gene predicted to a encode for a precursor containing at least 8 cysteine residues (S2 Table).

Similarly, homologs of *hypE* and *hypF* were detected in a gene cluster containing another gene encoding an ATP-grasp ligase. Genes encoding these proteins are typically part of the *hyp* operon, which is involved in the maturation of hydrogenase. Specifically, the 2 proteins cooperate to synthesize a thiocyanate ligand, which is transferred onto an iron center and used as a catalyst [67]. No other homologs of genes in the *hyp* operon were detected, however, suggesting that these protein-coding genes have adopted a novel function.

The remaining 18 families could not be grouped under a single denominator, nor could any single enzyme be found that clearly distinguished these groups as RiPP or non-RiPP BGCs. A wide variety of enzymes was found to be encoded by these gene clusters, including p450 oxidoreductases, flavoproteins, aminotransferases, methyltransferases, and phosphatases. In addition (and in line with features dominant in the positive training set), the predicted

precursor peptides were often rich in cysteine, serine, and threonine residues (S2 Table), which contain reactive hydroxyl and thiol moieties and are present in precursors of various known RiPP subclasses.

All candidate gene clusters presented here carry the features we selected, typical of RiPP BGCs: a low frequency of occurrence among the scanned genomes, a suitable precursor peptide, candidate modifying enzymes, transporters, regulators, and peptidases. However, many known RiPP BGCs were removed, suggesting that there may be more uncharacterized RiPP families among the gene clusters we discarded. While the complete dataset could not be covered here, the command-line application of decRiPPter has been set up to allow users to set their own filters. The pipeline can be run on any set of genomes. We recommend choosing a set of genomes that are sufficiently closely related to share a "core genome" for the COG-score calculations. At the same time, genomes should not be too similar, so that a wide variety of BGCs can be found among them that show variability in their presence/absence pattern across genomes. decRiPPter runs are visualized in an HTML output, in which the results can be further browsed and filtered by Pfam domains and other criteria, allowing users to find candidate families according to their preferences. The results from this analysis of the strict and the mild filter is available at https://decrippter.bioinformatics.nl.

## Discovery of a novel family of lanthipeptides

To validate the capacity of decRiPPter to find novel RiPP subclasses, we set out to experimentally characterize one of the candidate families (Fig 2; Other; red marker). Gene clusters belonging to this family shared several genes encoding flavoproteins, methyltransferases, oxidoreductases, and occasionally, a phosphotransferase. Importantly, the predicted precursor peptides encoded by these putative BGCs showed clear conservation of the N-terminal region, while varying more in the carboxyl-terminal region (S1 Text). This distinction is typical of RiPP precursors, as the N-terminal leader peptide is used as a recognition site for modifying enzymes, while the carboxyl-terminal core peptide can be more variable [20].

One of the gene clusters belonging to this candidate family was identified in *Streptomyces pristinaespiralis* ATCC 25468 (Fig 3A; Table 2). *S. pristinaespiralis* is known for the production of pristinamycin and was selected for experimental work since the strain is genetically tractable [68,69]. The gene cluster was named after its origin (*spr*: *Streptomyces pristinaespiralis* RiPP), and the genes were named after their putative function.

The gene cluster contains 4 genes encoding putative precursor peptides, although only 3 of the peptides (SprA1-A3) showed similarity to each other and to the other peptides in the same family (S1 Text). The fourth predicted precursor peptide (encoded by s*prX*) did not align with any of the other peptides and was assumed to be a false positive. The products encoded by *sprA1* and *sprA2* were highly similar to one another compared to the *sprA3* gene product (Fig 3A). Occurrence of 2 distinct genes for precursors within a single RiPP BGC is typical for 2-component lanthipeptides [70].

Most of the modifying enzymes present in the gene cluster had not previously been implicated in RiPP biosynthesis. The predicted *sprF2* gene product, however, shows high similarity to cysteine decarboxylases such as EpiD and CypD. These enzymes decarboxylate carboxyl-terminal cysteine residues, which is the first step in the formation of carboxyl-terminal loop structures called *S*-[(*Z*)-2-aminovinyl]-D-cysteine (AviCys) and *S*-[(*Z*)-2-aminovinyl]-(3S)-3-methyl-D-cysteine (AviMeCys) [71]. Several RiPP classes have been reported with this modification, including lanthipeptides, cypemycins, and thioviridamides, although they are only consistently present in cypemycins and thioviridamides. This type of modification is less common among lanthipeptides, with only 9 out of 120 lanthipeptide gene clusters in MIBiG

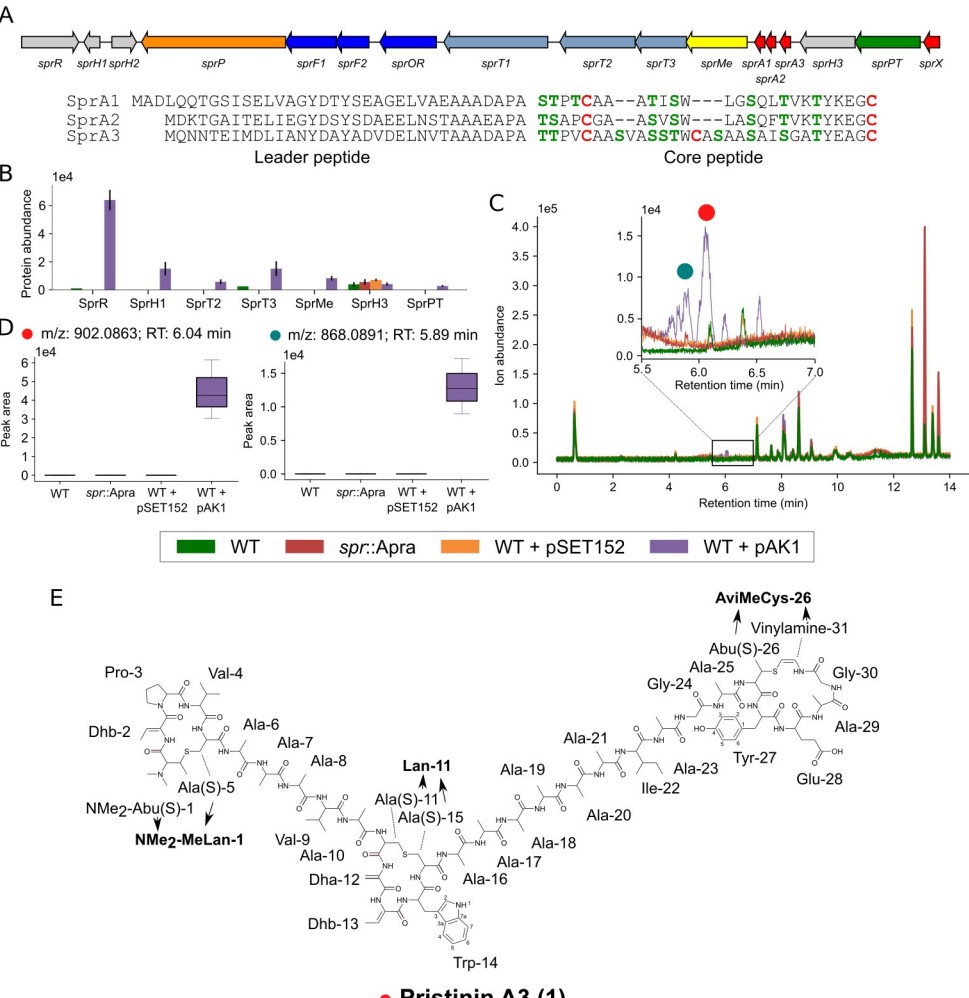

**Fig 3. The pristinin BGC (*spr*) of *S. pristinaespiralis* produces a highly modified RiPP.** (A) The *spr* gene cluster encodes 3 putative RiPP precursors, 3 transporters, a peptidase, and an assortment of modifying enzymes (see Table 2). Alignment of the predicted precursor peptides is given below. (B) Protein abundance of the products of the *spr* gene cluster in *S. pristinaespiralis* ATCC 25468 and its derivatives. Strains were grown in NMMP and samples were taken after 7 days. Enhanced expression of the regulator (from construct pAK1) resulted in the partial activation of the gene cluster. Proteins that could not be detected are not illustrated. (C) Overlay chromatogram of crude extracts from strains grown under the same conditions as under (B), samples after 7 days. Several peaks were detected in the extract from the strain with expression construct pAK1 between 7 and 8 minutes. (D) Boxplot of 2 peaks detected only in the strain with pAK1. The 2 masses could be related to 2 of the 3 precursors peptides. (E) 2D structure of pristinin A3 (**1**), derived from the SprA3 precursor. The compound has a mass of 2,703.235 Da. Numerical data of B, C, and D is available in S3 Data. BGC, biosynthetic gene cluster; NMMP, NH4-based Minimal Medium with Phosphate; RiPP, ribosomally synthesized and post-translationally modified peptide.

encoding the required decarboxylase. Genes encoding cysteine-decarboxylating enzymes are also present in non-RiPP gene clusters (S3 Table) and are also associated with other metabolic pathways [72]. In theory, though, this BGC could have been detected using a bait-based approach using these genes as queries.

A more detailed comparison with the gene clusters in MIBiG showed that 2 more genes from the thioviridamide gene cluster were homologous to 2 genes encoding a predicted phosphotransferase (*sprPT*) and a hypothetical protein (*sprH3*), respectively. Taken together with the homologous cysteine decarboxylase, it appeared that our gene cluster was distantly related

**Table 2. Annotation of the pristinin BGC (*spr*) of *S. pristinaespiralis*.**

| Gene name | Accession | NCBI Annotation of the putative gene product | Protein domains found | Proposed function |
|---|---|---|---|---|
| *sprR* | ALC22061.1 | LuxR family transcriptional regulator | | Cluster-specific regulator |
| *sprH1* | ALC22062.1 | hypothetical protein | | Unknown |
| *sprH2* | ALC22063.1 | hypothetical protein | | Unknown |
| *sprP* | ALC22064.1 | Peptidase M16 domain-containing protein | PF00675 Insulinase PF05193 Peptidase M16 inactive domain | RiPP maturation protease |
| *sprF1* | ALC22065.1 | Flavoprotein | PF01636 Phosphotransferase | Cysteine decarboxylation |
| *sprF2* | ALC22066.1 | Flavoprotein | PF02441 Flavoprotein | Cysteine decarboxylation |
| *sprOR* | ALC22067.1 | 5,10-methylene tetrahydromethanopterin reductase | PF00291 Luciferase-like monooxygenase | Reduction of dehydroalanine and dehydrobutyric acid |
| *sprT1* | ALC22068.1 | ABC transporter ATP-binding protein | PF00005 ABC transporter PF00664 ABC transporter transmembrane region | Transport |
| *sprT2* | ALC22069.1 | ABC transporter | PF12698 ABC-2 family transporter protein | Transport |
| *sprT3* | ALC22070.1 | ABC transporter ATP-binding protein | PF00005 ABC transporter PF13732 Domain of unknown function (DUF4162) | Transport |
| *sprMe* | ALC22071.1 | carminomycin 4-O-methyltransferase | PF00891 O-methyltransferase domain | N-terminal methylation |
| *sprA1* | ALC22072.1 | hypothetical protein | | RiPP precursor |
| *sprA2* | ALC22073.1 | hypothetical protein | | RiPP precursor |
| *sprA3* | ALC22074.1 | hypothetical protein | | RiPP precursor |
| *sprH3* | ALC22075.1 | hypothetical protein | PF17914 HopA1 effector protein family | Dehydration/cyclization |
| *sprPT* | ALC22076.1 | hypothetical protein | PF01636 Phosphotransferase | Dehydration/cyclization |
| *sprX* | ALC22077.1 | hypothetical protein | | Unknown |

to the thioviridamide gene cluster [73]. Thioviridamide-like compounds are primarily known for their thioamide residues, for which a TfuA-associated YcaO is thought to be responsible [32,74]. However, a YcaO homolog was not encoded by the gene cluster, making it unlikely that this gene cluster should produce thioamide-containing RiPPs.

Two strains were created to help determine the natural product specified by the BGC. For the first strain, the entire gene cluster was replaced by an apramycin resistance cassette (aac3 (IV)) by homologous recombination with the pWHM3 vector [75]. Both flanking regions were cloned into this vector, creating the vector pAK3. Subsequent homologous recombination resulted in a strain where the gene cluster was replaced by the *aac3(IV)* gene called *spr*::apra (Materials and methods). In case the gene cluster was natively expressed, this strain should allow for easy identification of the natural product by comparative metabolomics. In the second approach, we sought to activate the BGC in case it was not natively expressed. To this end, we targeted the cluster-situated *luxR*-family transcriptional regulatory gene *sprR*. The *sprR* gene was expressed from the strong and constitutive *gapdh* promoter from *S. coelicolor* ($p_{gapdh}$) on the integrative vector pSET152 [76]. The resulting construct (pAK1) was transformed to *S. pristinaespiralis* by protoplast transformation.

To assess the expression of the gene cluster in the transformants, we analyzed changes in the global expression profiles in 2 days and 7 days old samples of NH4-based Minimal Medium with Phosphate (NMMP)-grown cultures using quantitative proteomics (Fig 3B). Aside from the regulator itself, 6 out of the 16 other proteins were detected in the strain containing expression construct pAK1, while only SprPT could be detected in the strain carrying the empty vector pSET152. SprPT was also detected in the proteome of spr::apra, however, indicating a false positive. In the wild-type strain, SprT3 and SprR were detected, but only in a single replicate and at a much lower level. Overall, these results suggest that under the chosen growth conditions the gene cluster was expressed at very low amounts in wild-type cells and was activated when the expression of the likely pathway-specific regulatory gene was enhanced. This makes spr a likely silent BGC under the conditions tested.

To see if a RiPP was produced, the same cultures used for proteomics were separated into mycelial biomass and supernatant. The biomass was extracted with methanol, while HP20 beads were added to the supernatants to adsorb secreted natural products. Analysis of the crude methanol extracts and the HP20 eluents with high-performance liquid chromatography–mass spectrometry (HPLC-MS) revealed several peaks eluting between 5.5 and 7 minutes in the methanol extracts (Fig 3C), which were not found in extracts from wild-type strain or the strain containing the empty vector. Feature detection with MZmine followed by statistical analysis with MetaboAnalyst revealed 7 unique peaks, with $m/z$ between 707.3534 and 918.0807 (S9 Fig). The isotope patterns of these peaks showed that 6 of the identified ions were triply charged. Careful analysis of adduct ions and looking for mass increases consistent with Na- or K-addition led to the conclusion that these peaks corresponded to the $[M+3H]^+$ adduct, suggesting monoisotopic masses in the range of 2,604.273 and 2,754.242 Da. The highest signal came from the compound with a monoisotopic mass of 2,703.245. Four of the other masses seemed to be related to this mass, as they were different in mass increments of 4, 14, or 16 Da (S4A Data). We therefore reasoned that this mass was the product of one of the precursor peptides, while others were incompletely processed peptides. Another mass of 2,601.2433 could not be directly linked to the mass of 2,703.245. This mass was nevertheless only detected in extracts of the strain harboring pAK1 (Fig 3D), suggesting it is the product of another precursor peptide, although it is unclear whether or not it was the final product.

To further verify that the identified masses indeed belonged to the RiPP precursors in our gene cluster, we first removed the apramycin resistance cassette from Spr::apra using the pUWLCRE vector [77], creating strain Δspr (Materials and methods). The expression construct pAK1 and an empty pSET152 vector were transformed to the spr null mutant. When these transformants were grown under the same conditions, the aforementioned peaks were not detected, further suggesting that they were products of this gene cluster (S10A Fig).

Most masses were detected in only low amounts. In order to resolve this, we created a similar construct as pAK1, but this time using the low-copy shuttle vector pHJL401 as the vector [78]. The plasmid pAK2 was introduced into S. pristinaespiralis and the transformants grown in NMMP for 7 days. Extraction of the mycelial biomass with methanol resulted in a higher abundance of the masses previously detected (S10B Fig). Consistent with the MS profiles of pAK1 transformants, also pAK2 transformants produced an abundant peak corresponding to a monoisotopic mass of 2,703.245 Da, as well as a second peak corresponding to a monoisotopic mass of 2,553.260 Da. Many more masses were detected, most of which could be related to one of these 2 masses, suggesting these are the final products, related to 2 distinct precursors (S10C and S10D Fig, S4A and S4B Data).

We then performed MS/MS analysis of the extracts of the pAK2 transformants to identify the metabolites. Building on the hypothesis that the abundant peaks corresponded to the final products of SprA1-A3, we used their peptide sequences to map the fragments. The

fragmentation pattern of the peak with a mass of 2,703.245 Da could indeed be assigned to the sprA3 precursor sequence, but only when several mass adjustments of −16 Da, −18 Da, +28 Da, and −46 Da were applied (S11A Fig, S4C Data). Similarly, fragments for the mass of 2,553.260 could be matched to the SprA2 precursor sequence considering the same mass adjustments (S11B Fig, S4D Data).

All the −18 and −16 Da adjustments were predicted on serine and threonine fragments. These mass differences are typical of dehydration (−18 Da) of the residues to dehydroalanine (Dha) and dehydrobutyrine (Dhb). Reduction of these dehydrated amino acids (+2 Da) would then give rise to alanine and butyric acid residues, a modification that has been reported for lanthipeptides [79]. A modification of +28 Da suggests a dual methylation among the 5 N-terminal residues, which is consistent with the methyltransferase SprMe that is encoded by the *spr* gene cluster. The loss of −46 Da could be attributed to the carboxyl-terminal cysteine. This mass difference correlates to oxidative decarboxylation, which is consistent with the cysteine decarboxylase SprF2 that is encoded by the cluster. The loss of −18 Da in a threonine residue close to the modified cysteine suggests the presence of an AviMeCys group at the carboxyl-terminal end of the peptide. The lack of fragments for the residues T$^{-18}$YEAGC$^{-46}$ further supports the presence of an AviMeCys-containing carboxyl-terminal ring.

Surprisingly, no fragments were found of the residues S$^{-18}$S$^{-18}$T$^{-18}$WC in the center of SprA3, or for the N-terminal [T$^{-18}$T$^{-18}$PVC]$^{+28}$ region. Considering the other modifications typical of lanthipeptides, and the likely presence of a thioether crosslink in the AviMeCys group, we hypothesized the presence of thioether crosslinks between the Dhbs and cysteines. To find further support for this hypothesis, we treated the purified product of SprA3 with iodoacetamide (IAA). Iodoacetamide alkylates free cysteines, while cysteines in thioether bridges remain unmodified [80]. In agreement with our hypothesis, treatment with iodoacetamide did not affect the observed masses, despite the presence of 3 cysteines in the peptide (S10E Fig).

To further ascertain the presence of the proposed modifications, we purified the peak corresponding to the product of SprA3 precursor. Since the products were not detected when cultures were grown in 500 mL cultures, we grew $100 \times 20$ mL cultures (2 L total) of a transformant harboring the expression plasmid pAK2. The culture was then extracted and the extract was subjected to a series of chromatographic fractionations, which resulted in the purification of pristinin A3 (**1**) (Materials and methods). The purified compound was dissolved in deuterated dimethylsulfoxide (DMSO-$d_6$) for nuclear magnetic resonance (NMR) analysis. Extensive purification allowed us to purify 1.1 mg of the compound. While the amount of material meant that the NMR signal was low, we could derive many key features of the peptide in the $^1$H NMR spectrum (S12 Fig, S13A Fig). The NH signals in the $^1$H NMR spectrum were very broad using DMSO-$d_6$ as solvent. We therefore changed to CD$_3$CN:H$_2$O 9:1 as the solvent, which showed very good NH signals for the recently identified similar peptide cacaoidin [81]. Indeed, sharper peaks and better heteronuclear multiple bond correlation (HMBC) signals could be observed (S14B Fig). Reanalysis of pristinin A3 (**1**) using liquid chromatography–mass spectrometry (LC-MS) showed that the compound was partially oxidized, i.e., a mixture of compounds was analyzed in the NMR run using CD$_3$CN:H$_2$O as a solvent (S4F Data). MS/MS fragmentation suggested that the oxidation occurred consistently in the center and N-terminal ring structures (S4G Data).

Combined analysis of the 2D correlation spectroscopy (COSY), total correlation spectroscopy (TOCSY), heteronuclear single quantum coherence (HSQC), HMBC, and nuclear overhauser effect spectroscopy (NOESY) NMR spectra obtained in DMSO-$d_6$ (S13 Fig, S15 Table) supported the proposed structure of pristinin A3 (**1**) (Fig 3E). In the 2D spectra, several spin systems were identified, which were consistent with the amino acid sequence of SprA3 and the

MS/MS fragmentation data (S12 Fig). These amino acid residues were 2 Val, 2 Gly, 1 Pro, 1 Trp, 1 Ile, 1 Tyr, 1 Glu, and multiple mostly overlapping Ala. Additionally, we identified spin systems consistent with the proposed modified amino acid residues. These were 2 Dhbs, 2 β-thioalanines (Ala(S)), 1 Dha, 1 β-thioaminobutyric acid (Abu(S)), and 1 aminovinyl group. Due to weak signals, we could not use the HSQC-TOCSY spectra to further support the identified residues. There was no clear evidence in the NMR spectra of the presence of Thr or Ser amino acid residues, which corroborated the hypothesis that all the Thr and Ser residues identified in SprA3 had been modified.

We next sought evidence for the connectivity of the identified amino acids. The connectivity of the amino acid residues through NMR could be readily established through the Hα-NH (i, i+1), Hβ-NH (i, i+1), and NH-NH (i, i+1) NOESY correlations. Based on this, the Avi-MeCys-containing carboxyl-terminal ring and its extension up to Ala-21 could be unambiguously established to be in accordance with the proposed structure through the MS/MS data (S12 Fig). Importantly, the same structural fragment could be clearly observed in the sample analyzed in CD$_3$CN:H$_2$O 9:1, supporting the observation from the MS/MS data that the oxidation of pristinin A3 (**1**) was in the rings closer to the N-terminus. The NMR data in CD$_3$CN:H$_2$O confirmed the sequence of Ala-25 up to Glu-28, because some of the Hα and NH signals for these residues, which were overlapping in DMSO-$d_6$, were well separated in CD$_3$CN:H$_2$O (S14 and S15 Figs, S4E Data). Additionally, HMBC correlations could be observed to the carboxyl group of Glu in CD$_3$CN:H$_2$O. The NOESY correlations in DMSO-$d_6$ further unambiguously confirmed the peptide sequence observed in MS/MS for Dhb-2 to Ala-10, Dha-12 to Dhb-13, and Trp14 to Ala-16 (S4E Data). The sequence of Ala-17 to Ala-20 had overlapping Hα and NH signals. However, the correlation pattern observed and the peak integration support a series of alanine residues to be the connection between Ala-16 and Ala-21, as was also indicated by the MS/MS data.

It was not possible to establish the connection between Dhb-13 and Trp-14 using NMR. At the same time, a Dha–Dhb sequence could be clearly established using NMR. The fact that Dha and Dhb are the products of modified Ser and Thr residues, respectively, and the fact that the only Ser–Thr sequence in the SprA3 precursor lies before Trp, inevitably means that the observed Dha–Dhb structural fragment is connected to Trp-14 and positioned as Dha12 and Dhb-13. Finally, the thioether crosslinks of the proposed N-terminal and center ring structures could not be completely resolved based on NMR data alone. This is because the $^1$H NMR resonance for a CH/CH$_2$ group attached to a sulfur atom should be around δ$_H$ 3 ppm, which is close to the area where the water signal in DMSO-$d_6$ (δ$_H$ 3.3 ppm) is suppressed in the NMR experiments. Water suppression greatly affected the smaller signals around this area. Nevertheless, we managed to establish and position Ala(S)-5 and Ala(S)-15, both of which have to be part of a thioether bond as proven through the IAA labeling experiment discussed earlier. This left only 1 residue in each of the 2 additional rings observed in MS/MS, which was not accounted for by NMR (S12 Fig). Based on this, an NMe$_2$-Abu(S)-1 and Ala(S)-11 could be proposed to form thioether bridges with Ala(S)-5 and Ala(S)-15, respectively, resulting in the formation of N,N-dimethyl-β-methyllanthionine (NMe$_2$-MeLan) and lanthionine (Lan) residues, respectively. As a further evidence, we hydrolyzed the purified peptide with 6 M HCl at 110°C for 24 h. Under these conditions, the amide bond should be hydrolyzed, while the thioether bond should be unaffected [82]. The resulting mixture of amino acids was analyzed using LC-HRMS and was indeed found to contain peaks with exact masses corresponding to NMe$_2$-MeLan and Lan (S4H Data). Thus, the primary sequence of the peptide, the MS/MS fragmentation data, the NMR data, acid hydrolysis, and labeling experiments (Table 3) allowed us to elucidate the 2D structure of pristinin A3 (1) as shown in Fig 3E.

**Table 3. Summary of the different methods used to identify the amino acid residues of pristinin A3.**

| Amino acid residues | Gene sequence | HRMS/MS | NMR | Acid hydrolysis[a] | Amino acid residues | Gene sequence | HRMS/MS | NMR | Acid hydrolysis[a] |
|---|---|---|---|---|---|---|---|---|---|
| NMe₂-MeLan-1 = NMe₂-Abu(S)-1 + Ala(S)-5 | – | + | ±[b] | + | Ala-17 | – | + | + | + |
| Dhb-2 | – | + | + | – | Ala-18 | + | + | + | + |
| Pro-3 | + | + | + | + | Ala-19 | + | + | + | + |
| Val-4 | + | + | + | + | Ala-20 | – | + | + | + |
| Ala-6 | + | + | + | + | Ala-21 | + | + | + | + |
| Ala-7 | + | + | + | + | Ile-22 | + | + | + | + |
| Ala-8 | – | + | + | + | Ala-23 | – | + | + | + |
| Val-9 | + | + | + | + | Gly-24 | + | + | + | – |
| Ala-10 | + | + | + | + | Ala-25 | + | + | + | + |
| Lan-11 = Ala(S)-11 + Ala(S)-15 | – | + | ±[c] | + | AviMeCys-26 = Abu(S)-26 + Vinylamine-31 | – | + | + | – |
| Dha-12 | – | + | + | – | Tyr-27 | + | + | + | + |
| Dhb-13 | – | + | + | – | Glu-28 | + | + | + | + |
| Trp-14 | + | + | ±[d] | – | Ala-29 | + | + | + | + |
| Ala-16 | + | + | + | + | Gly-30 | + | + | + | – |

Symbols indicate whether residues and their connectivity were confirmed (+), partly confirmed (±), or not confirmed (−).

[a] Acid hydrolysis only confirms the amino residues, but not their connectivity.

[b] Only Ala(S)-5 could be observed in NMR.

[c] Only Ala(S)-15 could be observed in NMR.

[d] Trp-14 and its connectivity to Ala(S)-15 could be confirmed by NMR, but its connectivity to Dhb-13 could not be confirmed.

The RiPPs characterized here contain a number of modifications that have been previously identified in different other RiPPs. A recent study, which appeared around the time of submission of this paper, describes a RiPP found by activity-based screening called cacaoidin, which has many of the same modifications [81] and is additionally glycosylated. The serines converted to alanines in cacaoidin were all D-alanines. It therefore seems probable that the converted serines in pristinin A3 (**1**) were also converted to D-alanines, which could be determined by further chemical analyses. BLAST analysis shows that the genes of the cacaoidin BGC show low similarity to those in the *spr* BGC, and the precursor genes do not seem directly related. However, the same Pfam domains are found in both BGCs, indicating that both BGCs belong to the same RiPP class. The authors describing cacaoidin remark that these modifications were found previously in linaridins and lanthipeptides, and therefore named this class the lanthidins. While some enzymes encoded by the BGCs of this RiPP class indeed show low similarity to enzymes involved in the biosynthesis of characterized RiPPs, the combination of modifications makes it a novel RiPP subclass that was not previously detected by other RiPP genome mining tools. Overall, these findings further support the potential of decRiPPter to identify novel RiPP BGCs.

## The *sprH3/sprPT* gene pair is present in a wide variety of RiPP-like contexts

Taken together, we have shown that pristinin A3 contained a number of posttranslational modifications that are typical of lanthipeptides. The conversion of serine/threonine to alanine/butyric acid via reduction, the creation of an AviCys moiety, and the crosslinks to form

thioether bridges are all found in lanthipeptides and are dependent on dehydration of serine and threonine residues. Four different sets of enzymes called LanBC, LanM, LanKC, and LanL can catalyze these reactions in the biosynthesis of lanthipeptides and are used to designate the lanthipeptide class.

As stated before, no members of any of these enzyme families were found to be encoded by the gene cluster studied. However, *sprH3* and *sprPT* showed homology to 2 uncharacterized genes of the thioviridamide BGC. Thioviridamide contains an AviCys moiety, the formation of which requires a dehydrated serine residue. The enzymes responsible for dehydration and subsequent cyclization have not been identified yet [83,84]. Another RiPP subclass with an AviCys moiety is the linaridin subclass. Dehydration of the required serine is thought to be catalyzed by LinH or LinL, neither of which show similarity to the proteins encoded by the thioviridamide BGC or the *spr* BGC. Of note, the cacaoidin BGC also encoded 2 proteins with the same domains as SprH3 and SprPT (i.e., PF01636 and PF17914). Since the thioviridamide, cacaodin, and *spr* gene clusters share a common modification for which the enzyme is unknown, we hypothesize that SprH3 and SprPT carry out the dehydration and cyclization reactions and are therefore likely involved in the maturation of many different RiPPs, with dehydrated residues, AviCys moieties (as in thioviridamide), or thioether bridges (as in cacaoidin, SprA2, and SprA3). In the latter group, these enzymes candidate as core modifying enzymes of a new lanthipeptide subclass, which we designate lanthipeptide class V.

Lanthipeptide core modifying enzymes catalyze the most prominent reaction in lanthipeptide maturation and as such are present in many different genetic contexts [30,61]. To validate that SprH3 and SprPT are the sought-after modifying enzymes, we studied the distribution of the *SprH3/PT* gene pair across *Streptomyces* genomes analyzed by decRiPPter. Using CORASON [85] with the s*prPT* gene as a query yielded 195 homologs in various gene clusters (Fig 4, Materials and methods). The *sprPT/sprH3* gene pair was completely conserved across all gene clusters for which an uninterrupted contig of DNA was available, strongly supporting their functional interaction and joint involvement. Using the *sprH3* gene as a query yielded similar results. A total of 391 orthologs of the gene pair were found outside *Streptomyces*, particularly in Actinobacteria (219) and Firmicutes (161; S16 Fig). Distantly similar homologs of the gene pair were also identified in Cyanobacteria, Planctomycetes, and Proteobacteria.

Among the 195 identified gene clusters in *Streptomyces*, the majority (131) overlapped with a gene cluster detected by decRiPPter, indicating that the gene pair was within short intergenetic distance from predicted precursor gene in the same strand orientation. A large fraction (80) also passed the strictest filtering (Table 1), showing that among these gene clusters were many encoding biosynthetic machinery, peptidases, and regulators. In contrast, only 9 of the gene clusters overlapped with a BGC identified by antiSMASH. Four of these showed the gene pair in apparent operative linkage with a bacteriocin gene cluster, marked as such by the presence of a DUF692 domain. This domain is often associated with small prepeptides, such as the precursor peptides of methanobactin. Another 4 gene clusters detected by decRiPPter were only overlapping due to the gene pair being on the edge of a neighboring gene cluster.

The genetic context of the gene pairs showed a wide variation (Fig 4, right side). While some gene clusters were mostly homologous to the *spr* gene cluster (Fig 4, groups g and h), others shared only a few genes (groups a and d), and some only shared the gene pair itself (groups b, c, and e) (Table 4). Many other predicted enzyme families were found to be encoded inside these gene clusters, including YcaO-like proteins, glycosyltransferases, sulfotransferases, and aminotransferases. The large variation in genetic contexts combined with the clear association with a predicted precursor indicates that this gene pair likely plays a role in many different RiPP-associated genetic contexts, supporting their proposed role as a core gene pair. We emphasize, however, that not all of these BGCs necessarily specify lanthipeptides. Assuming

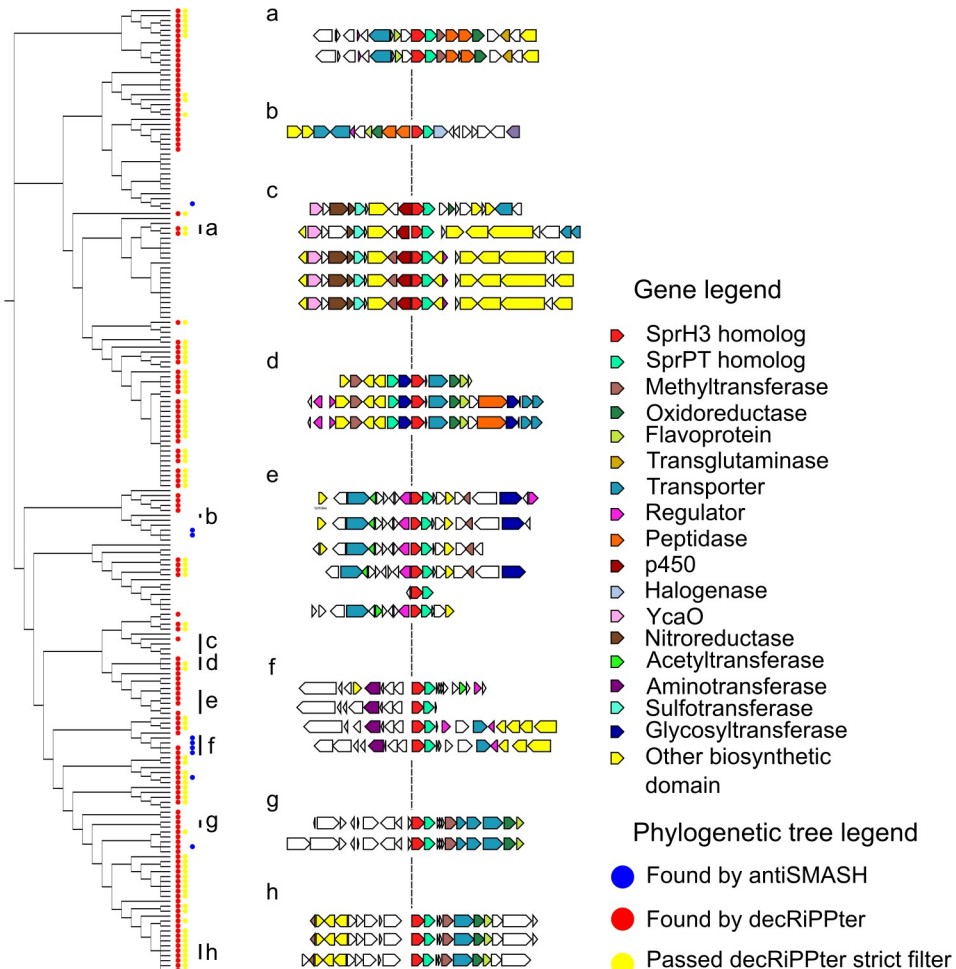

**Fig 4. Orthologs of *sprPT* and *sprH3* co-occur in a wide variety of genetic contexts.** (Left side) Phylogenetic tree of gene clusters containing homologs of *sprPT* and *sprH3*, visualized by CORASON. A red dot indicates that the genes were present in a gene cluster found by decRiPPter, a yellow dot that it passed the strict filter (Table 1). A blue dot indicates overlap with a BGC identified by antiSMASH. (Right side) Several gene clusters with varying genetic contexts are displayed. Group (g) represents the query gene cluster. The genetic context varies, while the gene pair itself is conserved. Color indicates predicted enzymatic activity of the gene products as described in the legend. The Newick file can be found in S3 Data. BGC, biosynthetic gene cluster; CORASON, CORe Analysis of Syntenic Orthologs to prioritize Natural Product-Biosynthetic Gene Clusters; decRiPPter, Data-driven Exploratory Class-independent RiPP TrackER.

that the proposed role for the products of *sprH3/PT* in dehydration of serine and threonine residues is correct, these modifications could also lead to AviCys moieties, such as in thioviridamide-like products, or simply remain dehydrated residues without the formation of a thioether bond. Further genetic and biochemical elucidation of the role of these enzymes is necessary to completely determine the scope of their reactions.

Furthermore, we searched for genes encoding enzymes whose functions are dependent on a lanthipeptide dehydration in their substrate, to find if they were associated with the *sprPT/sprH3* gene pair. Both within and outside *Streptomyces*, homologs of *sprF1* and *sprF2* were often found associated with the gene pair (*sprF1*: 251/586; 40.1%; *sprF2*: 281/586; 48.0%; S19 Table). Another modification dependent on the presence of dehydrated serine and threonine residues is the conversion of these to alanine and butyric acid, respectively. This conversion is

**Table 4. Co-occurrence of genes found in the *spr* gene cluster with homologs of *sprPT* in the analyzed 1,295 *Streptomyces* strains.**

| Gene name | Co-occurrence with s*prPT* (percentage) |
| --- | --- |
| *sprH3* | 99.49 |
| *sprMe* | 20 |
| *sprT1* | 35.38 |
| *sprT2* | 12.31 |
| *sprT3* | 12.82 |
| *sprOR* | 64.62 |
| *sprF1* | 39.5 |
| *sprF2* | 68.72 |
| *sprP* | 38.5 |
| *sprH1* | 9.0 |
| *sprH2* | 2.0 |
| *sprR* | 28.5 |
| *sprA1* | 1.03 |
| *sprA2* | 1.03 |
| *sprA3* | 16.92 |

catalyzed either by a zinc-dependent dehydrogenase (LanJ$_A$, also known as as LtnJ) or an NAD(P)H-dependent FMN reductase family enzyme (LanJ$_B$, also known as CrnJ) in lanthipeptides [79]. Outside *Streptomyces*, the genomic surroundings of the *sprPT/sprH3* gene pair occasionally contained homologs of the *lanj$_A$* gene (40/391; 10.1%). An example of such a BGC is that of pediocin A, which is a known antimicrobial compound, although its structure has yet to be resolved to the best of our knowledge [86]. These gene associations further imply that the sprH3/spr*PT* gene products carry out the canonical dehydration reactions.

A similar modification was observed for SprA2 and SprA3, despite that no homologs of the genes encoding LanJ$_A$ or LanJ$_B$ were identified within the *spr* gene cluster. However, *sprOR* encodes a putative oxidoreductase, and thus is a candidate for this modification. Supporting this, orthologs of *sprOR* were found frequently associated with either canonical lanthipeptide BGCs or the *sprPT/sprH3* gene pair (lanthipeptide: 124/462; *sprPT/sprH3*: 137/462; S4 Table). One of these lanthipeptide BGCs showed high homology to the lacticin 3147 BGCs from *Lactococcus lactis*. Lacticin 3147 contains several D-alanine residues as a result of conversion of dehydrated serine residues [87]. While all the genes, including the precursors, were well conserved between the 2 gene clusters, the *ltnJ* gene had been replaced by an *sprOR* homolog, suggesting that their gene products catalyze similar functions (S17 Fig). A recent paper describes the product of a BGC that contains a gene that similarly encodes a luciferase-like monooxygenase and shows that serine residues are indeed converted to alanine residues [30], further suggesting that this enzyme is responsible for this modification.

## Conclusion and final perspectives

The continued expansion of available genomic sequence data has allowed for discovery of large reservoirs of natural product BGCs, fueled by sophisticated genome mining methods. These methods must make tradeoffs between novelty and accuracy [11]. Tools primarily aimed at accuracy reliably discover large numbers of known natural product BGCs but are limited by specific genetical markers. On the other hand, while tools aimed at novelty may lead to the discovery of new natural products, these tools have to sacrifice on accuracy, resulting in a larger amount of false positives.

Here, we take a new approach to natural product genome mining, aimed specifically at the discovery of novel types of RiPPs. To this end, we built decRiPPter, an integrative approach to RiPP genome mining, based on general features of RiPP BGCs rather than selective presence of specific types of enzymes and domains. To increase the accuracy of our methods, we base detection of the RiPP BGCs on the one thing all RiPP BGCs have in common: a gene encoding a precursor peptide. With this method, we identify 42 candidate novel RiPP families, mined from only 1,295 *Streptomyces* genomes. These families are undetected by antiSMASH and show no clear markers identifying them as belonging to previously known RiPP BGC classes. While the approach to RiPP genome mining taken here inevitably gives rise to a higher number of false positives, we feel that such a "low-confidence/high novelty" approach [11] is necessary for the discovery of completely novel RiPP families. Additionally, users are able to set their own filters for the identified gene clusters, allowing them to search candidate RiPP families containing specific enzymes or enzyme types within a much more confined search space compared to manual genome browsing.

The product of one of the candidate classes was characterized as the first member of a new class of lanthipeptides (termed "class V") that was not detected by any other RiPP genome mining tool. Variants of this gene cluster are widespread across *Streptomyces* species, further expanding one of the most widely studied RiPP families. In addition, 2 proposed core genes were used to expand the family by finding additional homologs in *Actinobacteria* and *Firmicutes*. Taken together, this work shows that known RiPP families only cover part of the complete genomic landscape and that many more RiPP families likely remain to be discovered, especially when expanding the search space to the broader bacterial tree of life.

## Materials and methods

### decRiPPter pipeline

**Genome data preparation.** As input, decRiPPter uses a set of genomes from species that are part of the same taxonomic group (e.g., genus or family), which it requires for its comparative genomic analyses. decRiPPter downloads genomes from NCBI [88] based on NCBI taxonomic identifiers of species, genera, or higher orders of classification. Additional requirements for level of assembly (e.g., "Representative genome") can also be given. decRiPPter can reannotate genomes with prodigal 2.6.3 [89] and automatically does so when DNA FASTA files are given as input. In addition, users may analyze their own genomes, in isolation or in conjunction with downloaded genomes.

**SVM.** To predict RiPP gene clusters, we first collected positively and negatively labeled training data. The positive training data were collected from MIBiG [37] and recent literature, resulting in 175 RiPP precursors across 10 classes. For the negative training set, we generated a set of 20,000 short non-precursor sequences. Half of these were randomly selected from a set of 35,000 short proteins (<175 amino acids long) from Uniprot (queried June 2014) that were not similar to RiPP precursors based on an NCBI blastp search. The other half were randomly selected from a set of 17,000 translated intergenic sequences between a stop codon and the next start codon of sizes 30 to 300 nt taken from 10 genomes across the bacterial tree of life: *Escherichia coli*, *Bacillus subtilis*, *Streptomyces coelicolor*, *Bacteroides fragilis*, *Rhizobium etli*, *Chloroflexus aurantiacus*, *Synechococcus* sp. PCC 7002, *Opitutus terrae*, *Acidobacterium capsulatum*, and *Pirellula staleyi*. For all sequences from both the positive and negative training sets, we computed several physiochemical properties, such as its length, hydrophobicity, charge, counts of canonical amino acid residues and classes of amino acids, and highest counts of, e.g., cysteines and serines within contiguous blocks of 20 or 30 amino acids. All training data and

data collection scripts are available online (https://zenodo.org/record/3834818#.X7JmIOTsbvs).

We then utilized Scikit-Learn implementations of several different supervised machine-learning algorithms. We varied several parameters associated with a given algorithm (e.g., different kernel functions, a range of different values for penalty parameters, and different penalty functions). Furthermore, we mapped the accuracy as a function of scaling the dataset or changing class weights to take into account the unbalanced dataset (only approximately 1% of gene clusters in our dataset represent known RiPPs). The RiPP cluster classification accuracy of each combination of scaling, algorithm, and the corresponding set of parameters was evaluated using accuracy and area under receiver operating characteristics (ROC) curve and leave-one-class-out cross-validation. The final decRiPPter classifier uses an average of 57 SVMs to calculate precursor prediction. SVMs with three different kernel functions were trained: two with polynomal kernel function (SVM3: 3rd degree, coef0 of 2.154, kernel coefficient gamma of $2.78^*10^{-2}$, regularization parameter C of 0.158; SVM4: 4th degree, coef0 of 2.154, kernel coefficient gamma of $4.64^*10^{-3}$, regularization parameter C of 25.119) and one with a radial basis function kernel (SVMr: kernel coefficient gamma of $1^*10^{-5}$, regularization parameter C of $6.310^*10^5$). For each of these three types of SVMs, one SVM was trained with all training data, while eighteen more were trained by leaving out the sequences of one RiPP subclass from the positive training data at a time. The average of the scores obtained from all SVMs is taken as the final SVM score.

**COG scores calculation.** To calculate the relative frequency of occurrence of each gene, we constructed a pipeline to find all groups of homologous genes (S2 Fig). In the first step, protein-coding genes for which orthology can confidently be assigned are grouped into COGs. All proteins are aligned to one another using DIAMOND [46], and all bidirectional best hits (BBHs) are identified that share at least 60.0% similarity (S2A Fig). We established 2 requirements for genes to be confidently annotated as orthologs, based on recent papers [45,47]: (1) they should constitute BBHs; and (2) their immediate genomic surroundings should be conserved, i.e., the 2 flanking genes should also be BBHs between the 2 genomes. Genes fulfilling these 2 criteria are paired together, resulting in groups of orthologous genes. Among these groups, decRiPPter then selects those that are completely conserved across all genomes: Each group should contain at least 1 ortholog in each genome, and all orthologs in the group should all fulfill the same requirements for each genome pair. These groups are considered trueCOGs (S2B Fig).

In the second step, a cutoff for protein-coding gene sequence identity is determined for each genome pair, in order to separate orthologs as well as recently evolved paralogs from more distantly related homologs. For any given pair of genomes, the distribution of sequence identities of all gene pairs of their trueCOGs is calculated. The cutoff is then calculated as the average percentage identity, minus 3 times the standard deviation (S2C Fig). Any 2 aligned genes with a percentage identity higher than this cutoff are considered to be functionally closely related to one another and paired up. The resulting groups of homologous genes were clustered with the Markov Cluster Algorithm [48,49] (S2D Fig). From these groups, the relative frequency of occurrence of groups of homologous genes across all query genomes is calculated, called the COG-score (S2E Fig).

In cases when insufficient numbers of trueCOGs ($< = 10$) could be found in our analyses (because the set of genomes was too diverse and/or contained too many draft genomes that each miss some of the trueCOGs), the genomes were rearranged into smaller subgroups. We used 2 general rules to create the groups: (1) Groups should be as large as possible, so that trueCOGs found are conserved across many species, and represent conserved widespread genes. (2) Genomes should be compared to as many other genomes as possible, so as not to introduce

bias into the calculation of the COG-score. To fulfill both requirements, partially overlapping subgroups were formed, with the goal of letting each genome be a part of a collection of subgroups that together covered as many of the genomes as possible. To form the subgroups, a pair of genomes with the highest number of trueCOGs was used as a seed, and genomes were added one at a time until the number of trueCOGs dropped below the set cutoff. All the genomes in the group were said to be linked together by this group. The process of group formation was then repeated, starting with genomes for which no group had yet been formed. If all genomes were already part of at least 1 subgroup, the genomes were selected which were linked to the fewest genomes via the groups they were part of. The process was terminated when adding additional groups did not increase the number of links between genomes for several successive iterations.

**Gene cluster formation.** In this stage, decRiPPter identifies putative operon-like gene clusters around each candidate precursor peptide-encoding gene, by either of 2 different methods (S1 Fig): In the first method, called the simple method, genes in the same strand orientation as the candidate precursor peptide-encoding gene are added to the putative gene cluster if the intergenic distance to the previous gene is within a given cutoff. The second method, called the island method, uses both intergenic distance and levels of conservation (COG-score) to determine the gene clusters. First, all genes in the same strand orientation within 750 nucleotides of one another are identified and then grouped into islands. Within islands, genes should be almost directly adjacent (intergenic distance: $< = 50$ nucleotides). We then fused the islands together using the COG-scores (see above), building on the assumption that genes in a gene cluster should all have similar levels of conservation. Islands were fused together if the average of their COG-scores was within a set range (0.1 plus the sum of the standard deviations of both islands). Not all gene families have similar COG scores when they occur within the gene clusters thus formed; e.g., genes encoding ABC-transporters frequently have close relatives in other biomolecular systems and therefore often have higher COG scores. Hence, to counteract gene cluster formation breaking off prematurely, up to 2 outlier genes are allowed when fusing islands, if, after adding the outliers, more islands can be added that are within the range for COG-score deviation. Intergenic distances and cutoffs were iteratively fine-tuned to ensure gene clusters in known RiPP BGCs would be effectively found. Finally, gene clusters that overlap or lie within 50 nucleotides of one another are fused together.

**Annotation.** For purposes of data exploration (annotation and visualization), each gene cluster is extended to include the 5 flanking genes on either side, and all encoded proteins in the extended gene clusters are annotated with Pfam 31.0 [39] and TIGRFAM [40]. Lists were compiled of all TIGRFAM and Pfam domains associated with either peptidases, transporters, or regulators, using a combination of keyword searches on the Pfam and TIGRFAM websites, combined with manual curation. A list of protein domains associated with biosynthetic activity was constructed by linking Pfam domains to E.C. numbers using InterPro mappings [38]. Biosynthetic TIGRFAM domains were taken directly from the database. Each domain linked to an E.C. number was assumed to have enzymatic activity. The biosynthetic domain list was further expanded with domains used in the ClusterFinder [16] algorithm that were indicative of a biosynthetic gene cluster. The resulting lists are used by decRiPPter to mark proteins either as a regulator, peptidase, transporter, or biosynthetic enzyme, in that order, by seeing if any of the identified domains overlapped with the domains in the precompiled lists (S2 Data).

**Clustering.** To cluster the detected gene clusters, the distance between them is calculated in 2 different ways: (1) amino acid sequences of candidate precursor peptide-encoding genes in the gene clusters are aligned with NCBI BLAST[84] blastp (cutoff: 30 bitscore); and (2) the content of the gene clusters is compared by calculating the Jaccard index of their constituent protein domains (cutoff: 0.5). Gene clusters are paired only if they are paired by both methods.

The distance between paired gene clusters is calculated as the average between the Jaccard index and the percentage identity of the aligned precursors. Finally, pairs are clustered using MCL.

**Overlap with antiSMASH.** Overlap with antiSMASH was determined using antiSMASH 4.0 [54] run in minimal mode.

**Availability.** The decRiPPter pipeline is available at https://github.com/Alexamk/decRiPPter/. Data from the analysis discussed here are available at https://decrippter.bioinformatics.nl.

## Experimental

**Bacterial strain and growth conditions.** *Streptomyces pristinaespiralis* ATCC 25468 was purchased from DSMZ (DSM number 40338). Media components were purchased from Thermo Fisher Scientific, Sigma-Aldrich, or Duchefa Biochemie. For strain cultivation on solid media, *Streptomyces* spores were spread on mannitol soya flour agar (SFM; 20 g/L Agar, 20 g/L mannitol, 20 g/L soya flour, supplemented with tap water) prepared as described previously [90] and incubated at 30˚C. Spores were harvested after 4 to 7 days of growth when the strain started to produce a gray pigment by adding water directly to the plate and releasing the spores with a cotton swab. Spores were centrifuged and stored in 20% glycerol.

For cultivation in liquid media, 20 to 50 μL of a dense spore stock was inoculated into 100 mL shake flasks with coiled coils containing 20 mL of the medium of interest. For extractions, NMMP was used (0.60 mg/L MgSO$_4$, 5 mg/L NH$_4$SO$_4$, 5 g/L Bacto casaminoacids, 1 mL trace elements (1 g/L ZnSO$_4$.7H$_2$O, 1 g/L FeSO$_4$.7H$_2$O, 1 g/L MnCl$_2$.4H$_2$O, 1 g/L CaCl$_2$, anhydrous)), while for genomic DNA isolation, a 1:1 mixture of TSBS: YEME with 0.5% glycine and 5 mM MgCl$_2$ was used (TSBS: 30 g/L Bacto Tryptic Soy Broth, 100 g/L sucrose; YEME: Bacto Yeast Extract: 3 g/L, Bacto Peptone 5 g/L, Bacto Malt Extract 3 g/L, glucose 10 g/L, sucrose 340 g/L).

*E. coli* strains JM109 and ET8 were used for general cloning purposes and demethylation, respectively. Strains were cultivated in liquid LB and on LB-agar plates at 37˚C.

**Molecular biology.** All materials and primers were purchased from Sigma-Aldrich or Thermo Fisher Scientific unless stated otherwise. Restriction enzymes and T4 ligase were purchased from NEB. Restriction and ligation protocols were followed as per manufacturer's description. For amplification of DNA fragments with PCR, Pfu polymerase was used. Primers were designed with T$_m$ of the annealing region roughly equal to 60˚C. Standard PCR protocols consisted of 30 cycles (45-second DNA melting @ 95˚C, 45-second primer annealing @ 55˚C to 65˚C, 60-second to 180-second primer elongation @ 72˚C), but PCR protocols were optimized where necessary.

*S. pristinaespiralis Spr::Ap* deletion mutants were created by replacing the gene cluster with an *aac(3)IV* apramycin resistance cassette via homologous recombination. The −1507/−39 and +135/+1641 regions upstream and downstream of the cluster were amplified by PCR with the spr_LF_F/spr_LF_R and spr_RF_F/spr_RF_R primer pairs, respectively, and inserted into the pWHM3-oriT vector into the EcoRI/HindIII sites (S5 Data). The *aac(3)IV* apramycin resistance cassette was inserted into the XbaI site, creating pAK3. pAK3 was transformed to *E. coli ET8* for DNA demethylation, purified, and transformed to *S. pristinaespiralis* by protoplast transformation. Transformation mixtures were plated out on R5, prepared as described earlier [90]. After 14 to 18 hours, the plates were overlaid with 1.2 mL H$_2$O containing 10 μg thiostrepton and 25 μg apramycin. Three colonies were picked after 4 days of growth and spread onto SFM plates without added antibiotic to allow for homologous recombination. Colonies containing the correct phenotype (apramycin-resistant and thiostrepton-sensitive) were

picked, and the homologous recombination was confirmed by PCR, using the spr_del_-check_F/spr_del_check_R primer pair.

Constructs for the overexpression of the *sprR* regulator were constructed as follows: The *sprR* gene was amplified from the genomic DNA of *S. pristinaespiralis* using the sprR_F/sprR_R primer pair and placed into the EcoRI/XbaI site of the pSET152 vector. The −0/−457 upstream region of glyceraldehyde 3-phosphate dehydrogenase amplified from the genome of *S. coelicolor* was obtained from previous studies [91,92] and inserted into the EcoRI site and the engineered NdeI site, placing it directly upstream of the *sprR* gene. To create vector pAK2, the entire region between the EcoRI and XbaI sites was excised and inserted into the pHJL401 vector.

**Extractions.** Strains were cultured in 100 mL shake flasks containing 20 mL NMMP, with coiled coils at 30°C for 7 days. A volume of 20 μg/mL thiostrepton was added to cultivate strains containing pHJL401. Mycelium was collected by centrifugation, washed twice with sterile MiliQ water, and extracted with 5 mL methanol by shaking overnight at 4°C. The methanol was collected and centrifuged at 4°C to clear it of cellular debris and precipitates. The crude extracts were dried and weighed, and dissolved in methanol at a concentration of 1 mg/mL for further analysis.

**Peptide purification.** For large-scale extraction, the strain proved incapable of producing the desired compound when grown in large shake flasks. Therefore, 2 L NMMP prepared as above was inoculated with 2.5 mL of a dense spore stock *S. pristinaespiralis* with pAK3 and split over one hundred 100 mL shake flasks. Thiostrepton was added as described above. The cultures were grown for 14 days, pooled together, and extracted with an equivalent volume of butanol. The butanol extracted was then evaporated in vacuo to yield 1.7 g of crude extract.

The resulting crude extract was adsorbed on silica gel 60 (40 to 60 μm, Sigma Aldrich) and dry loaded on a vacuum liquid chromatography (VLC) column (3 × 30 cm) packed with the same material. The column was eluted with 200 mL fractions of a gradient comprised of (v/v): hexane, hexane–EtAc (1:1), EtAc, EtAc-MeOH (3:1), EtAc-MeOH (1:1), EtAc–MeOH (1:3), and finally MeOH. The fractions containing the compound of interest were pooled, concentrated, and further purified using Waters preparative HPLC system comprised of 1525 pump, 2707 autosampler, and 2998 PDA detector. The pooled fraction (112.9 mg) was injected into a SunFire $C_{18}$ column (10 μm, 100 Å, 19 × 150 mm). The column was run at a flow rate of 12.0 mL/min, using solvent A (0.1% FA in $H_2O$) and solvent B (0.1% FA in ACN), and a gradient of 30% to 60% B over 20 minutes. HPLC purification was monitored at 254 nm, and eventually resulted in compound **1** (1.1 mg).

**LC-MS analysis.** LC-MS/MS acquisition was performed using Shimadzu Nexera X2 UHPLC system, with attached photodiode array detector (PDA), coupled to Shimadzu 9030 QTOF mass spectrometer, equipped with a standard electrospray ionization (ESI) source unit, in which a calibrant delivery system (CDS) is installed. The dry extracts were dissolved in MeOH to a final concentration of 1 mg/mL, and 2 μL were injected into a Waters Acquity Peptide BEH $C_{18}$ column (1.7 μm, 300 Å, 2.1 × 100 mm). The column was maintained at 40°C and run at a flow rate of 0.5 mL/min, using 0.1% formic acid in $H_2O$ as solvent A and 0.1% formic acid in acetonitrile as solvent B. A gradient was employed for chromatographic separation starting at 5% B for 1 minute, then 5% to 85% B for 9 minutes, 85% to 100% B for 1 minute, and finally held at 100% B for 4 minutes. The column was re-equilibrated to 5% B for 3 minutes before the next run was started. The LC flow was switched to the waste the first 0.5 minute, then to the MS for 13.5 minutes, then back to the waste to the end of the run. The PDA acquisition was performed in the range 200 to 400 nm, at 4.2 Hz, with 1.2 nm slit width. The flow cell was maintained at 40°C.

The MS system was tuned using standard NaI solution (Shimadzu). The same solution was used to calibrate the system before starting. System suitability was checked by including a standard sample made of 5 μg/mL thiostrepton; which was analyzed regularly in between the batch of samples.

All the samples were analyzed in positive polarity, using data-dependent acquisition mode. In this regard, full scan MS spectra ($m/z$ 400 to 4,000, scan rate 20 Hz) were followed by 3 data-dependent MS/MS spectra ($m/z$ 400 to 4,000, scan rate 20 Hz) for the 3 most intense ions per scan. The ions were selected when they reach an intensity threshold of 1,000, isolated at the tuning file Q1 resolution, fragmented using collision-induced dissociation (CID) with collision energy ramp (CE 10 to 40 eV), and excluded for 0.05 second (1 MS scan) before being reselected for fragmentation. The parameters used for the ESI source were: interface voltage 4 kV, interface temperature 300˚C, nebulizing gas flow 3 L/min, and drying gas flow 10 L/min.

**LC-MS–based comparative metabolomics.** All raw data obtained from LC-MS analysis were converted to mzXML centroid files using Shimadzu LabSolutions Postrun Analysis. The converted files were imported and processed MZmine 2.5.3 [93]. Throughout the analysis, $m/z$ tolerance was set to 0.002 $m/z$ or 10.0 ppm, RT tolerance was set to 0.05 minute, noise level was set to 2.0E2, and minimum absolute intensity was set to 5.0E2 unless specified otherwise. Features were detected (polarity: positive, mass detector: centroid), and their chromatograms were built using the ADAP chromatogram builder [94] (minimum group size in number of scans: 10; group intensity threshold: 2.0E2). The detected peaks were smoothed (filter width: 9), and the chromatograms were deconvoluted (algorithm: local minimum search; chromatographic threshold: 90%; search minimum in RT range: 0.05; minimum relative height: 1%; minimum ratio of peak top/edge: 2; peak duration 0.03 to 3.00 minute). The detected peaks were deisotoped (maximum charge: 5; representative isotope: lowest $m/z$). Peak lists from different extracts were aligned (weight for RT = weight for $m/z$; compare isotopic pattern with a minimum score of 50%). Missing peaks detected in at least one of the sample were filled with the gap filling algorithm (RT tolerance: 0.1 minute). Among the peaks, we identified fragments (maximum fragment peak height: 50%), adducts ([M+Na]$^+$, [M+K]$^+$, [M+NH$_4$], maximum relative adduct peak height: 3,000%) and complexes (ionization method: [M+H]$^+$, maximum complex height: 50%). Duplicate peaks were filtered. Artifacts caused by detector ringing were removed ($m/z$ tolerance: 1.0 $m/z$ or 1,000.0 ppm), and the results were filtered down to the retention time of interest. The aligned peaks were exported to a MetaboAnalyst file. From here, peaks were additionally filtered to keep only peaks present in all 3 replicates, using in-house scripts. The resulting peak list was uploaded to MetaboAnalyst [95], log transformed, and normalized with Pareto scaling without prior filtering. Missing values were filled with half of the minimum positive value in the original data. Heatmaps and volcano plots were generated using default parameters.

**Mass spectrometry-based quantitative proteomics.** A volume of 20 μL of dense spore stocks were inoculated in NMMP and grown for 7 days as described above. A total of 1 mL samples were taken after 2 and 7 days. Mycelium was gathered by centrifugation and washed with disruption buffer (100 mM Tris-HCl (pH 7.6), 0.1 M dithiothreitol). The samples were sonicated for 5 minutes (in cycles off 5 seconds on, 5 seconds off) to disrupt the cell wall and centrifuged at max speed for 10 minutes to collect the proteins. Proteins were then precipitated using chloroform-methanol [96]. The dried proteins were dissolved in 0.1% RapiGest SF surfactant (Waters) at 95˚C. Protein digestion steps were done according to van Rooden and colleagues [97]. After digestion, formic acid was added for complete degradation and removal of RapiGest SF. Peptide solution containing 8 μg peptide was then cleaned and desalted using the STAGETipping technique [98]. Final peptide concentration was adjusted to 40 ng/μL with 3% acetonitrile and 0.5% formic acid solution. A total of 200 ng of digested peptide was injected

and analyzed by reversed-phase liquid chromatography on a nanoAcquity UPLC system (Waters) equipped with HSS-T3 C18 1.8 μm, 75 μm × 250 mm column (Waters). A gradient from 1% to 40% acetonitrile in 110 minutes was applied; [Glu$^1$]-fibrinopeptide B was used as lock mass compound and sampled every 30 seconds. Online MS/MS analysis was done using Synapt G2-Si HDMS mass spectrometer (Waters) with an UDMS$^E$ method set up as described [97].

Mass spectral data were generated using ProteinLynx Global SERVER (PLGS, version 3.0.3), with MS$^E$ processing parameters with charge 2 lock mass 785.8426 Da. A reference protein database was downloaded from GenBank with the accession number GCA_001278075.1. The resulting data were imported to ISOQuant [99] for label-free quantification. TOP3 quantification result from ISOQuant was used when further investigating the data.

**Iodoacetamide treatment.** Reaction mixtures were prepared based on earlier reported studies [80]. Reaction mixtures of 20 μL containing 0.25 mg/mL purified peptide, 13 mM TCEP, 25 mM IAA, and 250 mM HEPES (pH 8.0) in $H_2O$ were left at room temperature for 1 hour in the dark. Reaction mixtures were cleaned using the STAGETipping technique [98].

**Protein hydrolysis.** A total of 0.2 mg of purified peptide was dissolved in 3 mL 6 M HCl and sealed inside a glass ampule, based on earlier studies [100]. The mixture was heated to 110˚C for 24 hours. The HCl was removed by repeated drying and dissolving of the peptide with $H_2O$. The peptide was afterwards dissolved in 50 μL $H_2O$ and analyzed with LCMS as described above.

**NMR.** NMR data were recorded on Bruker Ascend 850 NMR spectrometer (Bruker BioSpin GmbH), equipped with a 5 mm cryoprobe. The sample was measured in a 3 mm NMR tube through the use of an adapter. All NMR experiments were performed with suppression of the water peak in the solvent.

## Data analysis

**Streptomyces analysis.** For genome mining of *Streptomyces*, we downloaded all available genomes falling under the taxonomic identifier for *Streptomyces* (1883). Genomes were reannotated with prodigal 2.6.3 and processed with the pipeline described above. Gene clusters were formed with the island method.

The results of the strict and the mild filter are available at https://decrippter.bioinformatics.nl.

Gene clusters passing the strict filter were further curated using the following general criteria:

1. If more than half of the genes appeared involved in a known metabolic pathway not related to RiPPs, the BGC was filtered. This includes many examples, including

   1. The LysW pathway;

   2. Sugar metabolism pathways;

   3. Cell wall modification pathways;

   4. DNA modification pathways.

   >In contrast, if only a few links to a known pathway could be identified, it seemed more likely to us that the candidate RiPP BGC was evolved from that pathway, such as in the MauE and MauD-encoding BGCs.

2. The predicted precursor itself was annotated as a biosynthetic enzyme (commonly ferredoxin, which is also rich in cysteine residues). This step is done automatically in the currently online version of decRiPPter.

3. The precursor was extracted from an intergenic region, but there appeared little space for a promoter region preceding it. This step may be automated in a later stage.

4. The BGC contained some abnormalities, such as extreme lengths for RiPP BGCs (>50 genes) or large intergenic gaps, or large genes (>10,000 aa)

**Genomic context analysis.** CORASON [85] was used with the number of flanking genes set to 15, on the *Streptomyces* genomes analyzed with the query of interest. Results were parsed using in-house scripts and compared to decRiPPter output. NCBI BLAST was used to find additional homologs of genes of interest within the clusters, with a cutoff of 30% ID similarity.

**Comparison with NeuRiPP and NLPPrecursor.** NeuRiPP classifications were performed using the parallel CNN network with the network weights provided by the author [34]. NLPPrecursor was installed and executed with default settings [35]. All open reading frames were analyzed with both methods, and completely overlapping precursor hits on the same frame were removed, as in the decRiPPter pipeline.

**Comparison with RODEO.** All positively scored precursor peptides were extracted from genome mining studies done with RODEO [26,27,29–31]. For lasso peptides, only the most recent dataset was used [29]. All precursor peptides were analyzed with decRiPPter's SVM, using a cutoff of 0.9. To determine BGC overlap, all genes for each of the core enzymes used as a query by RODEO within the 1,295 *Streptomyces* genomes analyzed in this study were extracted. For the study involving sactipeptides/ranthipeptides [26], only genes for radical SAM enzymes were extracted if a nearby precursor was detected, since decRiPPter is dependent on the presence of a nearby precursor to detect the BGC.

## Supporting information

**S1 Fig. decRiPPter forms putative gene clusters around candidate precursor peptide-encoding genes.** Two examples are provided here to illustrate identification of putative gene clusters in decRiPPter. (A) In the *sapB* gene cluster, 4 genes form the main BGC. These 4 genes are sequential, share the same strand orientation, and lie within a small distance of one another (< = 50 nt). They are therefore fused together into a single gene cluster. The flanking genes are on opposite strands and therefore not considered. (B) The *skfA* BGC consists of 8 genes sequential genes that share the same strand orientation. However, it is flanked by several other genes that also share the same strand orientation, within relatively short intergenic distances (< = 200 nucleotides). Using the island method, the genes are first fused into 6 islands, within 50 nucleotides distance of one another (indicated by lines underneath the genes). These islands may then be fused depending on the COG-score, which does not happen here because the difference is too large. The result is that the flanking genes, with a too high COG-score, are not added, and the correct BGC remains. BGC, biosynthetic gene cluster; COG, cluster of orthologous genes; decRiPPter, Data-driven Exploratory Class-independent RiPP TrackER. (PNG)

**S2 Fig. decRiPPter determines the frequencies of occurrence of genes to calculate the COG score.** In this example, the COG scores of 4 genomes are calculated. (A) All encoded proteins are aligned to find BBHs (edges). All clusters of BBHs conserved across all genomes are displayed as red. If 1 genome does not contain a homologous gene, or the gene in question is not a BBH with all genes from the cluster from other genomes, it is not considered a conserved group of BBHs. (B) If the flanking genes of the clusters of BBHs are also part of clusters of BBHs, the center genes are considered to form a trueCOG. Of the 3 cases displayed here, only the leftmost group passes this criterion; for the center group, not all genes are conserved; and for the right group, not all genes are BBHs with one another in the flanking groups. (C) The

genes in all trueCOGs between each genome pair are used to create a sequence identity cutoff to use for all protein-coding genes in a given pair of genomes. (D) All genes are paired using the sequence identity cutoffs determined in the previous step. (E) The COG-score is calculated for each gene. Typically, a bimodal distribution can be seen, with many genes either conserved across all genomes, or only present in a single organism. BBH, bidirectional best hit; COG, cluster of orthologous genes; decRiPPter, Data-driven Exploratory Class-independent RiPP TrackER; trueCOG, true cluster of orthologous genes.
(PNG)

**S3 Fig. COG and SVM scores in all analyzed 1,295 *Streptomyces* genomes. (**A) COG scores of all genes in all 1,295 analyzed *Streptomyces* genomes. A high COG score indicates presence of homologs in many different genomes, while a low COG score indicates a more infrequent distribution. COG scores were calculated as described in the methods. (B) Distribution of the scores assigned by decRiPPter's SVM classifier. A total of $7.1 * 10^7$ small ORFs were analyzed. Based on the slight enrichment of precursors with scores $> = 0.90$, the cutoff was set at 0.9. (C) Comparison of COG scores of antiSMASH-detected gene clusters. COG scores were averaged over all genes in the predicted gene clusters. COG scores averaged $0.311 \pm 0.249$ for all gene clusters, and $0.234 \pm 0.166$ for RiPP gene clusters. (D) Comparison of average COG scores of BUSCO genes. The average of each BUSCO [51,52] gene was calculated for each genome analyzed. Numerical data of figures A, B, C, and D are available in S3 Data. BUSCO, Benchmarking set of Universal Single-Copy Orthologs; COG, cluster of orthologous genes; ORF, open reading frame; RiPP, ribosomally synthesized and post-translationally modified peptide; SVM, Support Vector Machine.
(PNG)

**S4 Fig. COG-scores calculations depend on genome group size. (**A) As the minimum number of trueCOGs increases, the number of genomes that can be analyzed together (red line) decreases. In addition, the average COG cutoff (blue line) decreases when more trueCOGs are added, and the spread of COG cutoffs (shaded area; average cutoff ± the standard deviation) increases, suggesting that additional trueCOGs that were added were less conserved and showed higher variability in sequence similarity. (B) TrueCOG distribution between 36 randomly sampled genome pairs. Based on these distributions, COG cutoffs were determined. Numerical data of figures A and B are available in S3 Data. COG, cluster of orthologous genes; trueCOG, true cluster of orthologous genes.
(PNG)

**S5 Fig. Three machine-learning–based RiPP precursor classifiers give highly different results.** All small ORFs from the 1,295 *Streptomyces* genomes were classified by DeepRiPP's NLPPrecursor [35] module, NeuRiPP [34], and decRiPPter. The 3 tools have only a small overlap (10,691 hits). NLPPrecursor scored 6 times more hits as positive, and NeuRiPP roughly half when compared to decRiPPter. Many of these hits were very small ORFs (≤30 amino acids; (B)), though, while most of decRiPPters predicted precursors were larger than that. The exact accuracy of these tools cannot be determined, as it is unclear which of these hits are false positives, and which are hits in novel RiPP BGCs. BGC, biosynthetic gene cluster; ORF, open reading frame; RiPP, ribosomally synthesized and post-translationally modified peptide.
(PNG)

**S6 Fig. GC-content of randomly sampled Prodigal-detected precursor hits (A) and intergenic precursor hits (B).** GC content is shown as the moving average of the first, second, and third positions, using a window-size of 5 and a step-size of 2. Only a small percentage of intergenic hits showed clear distinction between the 3 moving averages as in the Prodigal-detected

hits, suggesting the majority of these are not encoding genes. Numerical data of figures A and B are available in S3 Data. GC, guanine-cytosine.
(PNG)

**S7 Fig. Gene cluster formation effectively covers antiSMASH and MIBiG BGC core gene sections.** In the simple gene cluster formation method, genes are sequentially added as long as they are in the same strand orientation, within a certain distance. At a distance of 700 nucleotides, all MIBiG core gene sections are covered (A), as well as 91% (3947/4321) of antiSMASH core gene sections. (B). In the "island method," genes are first fused into islands, which may be further fused if their average COG-scores are within a cutoff. Using just the standard deviation of the islands as a cutoff resulted in incomplete coverage of both the MIBiG and the antiSMASH core sections (C, D, middle boxes). Increasing the cutoff to the standard deviation plus 0.1 resulted in comparable coverage (C, D, right boxes) of these sections when compared to the simple method (C, D, left boxes). In addition, the overall gene cluster length (E) and variation of COG scores (F) within all formed gene clusters decreased. Numerical data of figures A, B, C, D, E, and F are available in S3 Data. BGC, biosynthetic gene cluster; COG, cluster of orthologous genes; MIBiG, Minimum Information about a Biosynthetic Gene cluster.
(PNG)

**S8 Fig. Combining precursor similarity with domain similarity is an effective strategy to group RiPP subclasses.** Starting at precursor similarity bitscore cutoffs of 20 and Jaccard scores of overlapping protein domains found in MIBiG RiPP BGCs of 0.4, the number of intraclass homologies is larger than the number of crossclass homologies. Combining the 2 methods greatly decreases the number of cross-class homologies found, proving it as an effective method to group RiPP BGCs of different subtypes. Numerical data of figures A, B, and C are available in S3 Data. BGC, biosynthetic gene cluster; MIBiG, Minimum Information about a Biosynthetic Gene cluster; RiPP, ribosomally synthesized and post-translationally modified peptide.
(PNG)

**S9 Fig. Heatmap of extracted peaks reveals 7 peaks that are uniquely observed in strains containing the expression construct pAK1.** Color of the area indicates a log-fold increase. Numerical data are available in S3 Data.
(PNG)

**S10 Fig. Chromatograms comparing the extracted compounds in knockout strains and highly producing strains. (**A) Strains lacking the *spr* gene cluster are unable to produce the extracted products, even when transformed with pAK1. (B) Chromatogram of methanol extracts made from *S. pristinaespiralis* harboring no vector (WT), an empty pHJL401 vector (pHJL401), or pAK2 (pHJL401 with *sprR* behind $p_{gap}$). A large peak can be seen in the extracts of strains harboring pAK2, not seen in extracts of the other strains. (C) Volcano plot comparing extracts of the strain containing pAK2 with the strain containing pHJL401. Peaks in pink had $p$-value $\geq 0.1$ and a fold-change of $\geq 2$. A large collection of peaks can be identified with log2(fold-change) $\geq 10$. The 2 largest peaks (bold) corresponding to different monoisotopic masses could be related to the SprA2 and SprA3 precursors by MS/MS (S11 Fig). Many of the other masses eluted at comparable times and had masses that were close to the 2 major peaks, suggesting they were derived from them. Clear mass differences could be identified for some of the identified masses (S4B Data). Whether the largest peaks indeed correspond to the final product remains to be determined. (D) Extracted ion chromatograms of the 2 major peaks identified from the volcano plot. The 2 masses were only detected in the strain harboring pAK2. (E) Labeling experiments with IAA provide further support for the proposed structure

of SprA3. (Purple) IAA covalently attaches to free sulfur groups of cysteines. However, the SprA3 peak was unaltered by IAA treatment, despite the presence of 3 cysteines in the peptide, strongly suggesting that these cysteines are not free. Numerical data of figures A, B, C, D, and E are available in S3 Data. IAA, iodoacetamide; MS/MS, tandem mass spectrometry; WT, wild-type.
(PNG)

**S11 Fig. Fragmentation patterns of 2 highly extracted peaks can be matched to the SprA2 and SprA3 precursors.** A full list of the modifications applied can be found in S4C and S4D Data. Numerical data of the chromatogram are available in S3 Data.
(PNG)

**S12 Fig. Key 2D NMR correlations observed for pristinin A3 (1).** No clear correlations could be observed for the red parts of the structure, which were confirmed through other techniques. Bold arrows are for correlations which were better observed in $CD_3CN:H_2O$ 9:1.
(TIF)

**S13 Fig. NMR spectra of pristinin A3 (850 MHz, in DMSO-*d6*, 298K).** (A) $^1$H NMR spectrum with water suppression. The peak at 3.17 ppm is due to traces of methanol in the sample. (B) $^1$H–$^1$H COSY spectrum. (C) 2D TOCSY spectrum. (D) Multiplicity-edited HSQC spectrum. (E) HSQC-TOCSY spectrum. (F) HMBC spectrum. (G) NOESY spectrum.
(PNG)

**S14 Fig. NMR spectra of pristinin A3 (850 MHz, in $CD_3CN:H_2O$ 9:1, 297 K, first run).** (A) $^1$H NMR spectrum with water suppression. (B) $^1$H–$^1$H COSY spectrum. (C) 2D TOCSY spectrum. (D) Multiplicity-edited HSQC spectrum. (E) HMBC spectrum.
(PNG)

**S15 Fig. NMR spectra of pristinin A3 (850 MHz, in $CD_3CN:H_2O$ 9:1, 297 K, second run).** (A) $^1$H NMR spectrum. (B) $^1$H–$^1$H COSY spectrum. (C) NOESY spectrum. (D) HMBC spectrum.
(PNG)

**S16 Fig. Homologs of the *sprPT* and *sprH3* gene pair are present outside *Streptomyces*.** Most homologs were found in *Actinobacteria* and *Firmicutes*, although a few additional candidates were found in *Proteobacteria*, *Cyanobacteria*, and *Planctomycetes*.
(PNG)

**S17 Fig. Comparison of the lacticin 3147-like gene cluster from *Lactococcus lactis* with a homologous cluster from *Streptomyces olivaceus*.** Genes encoding both precursors, both LanM-like modifying enzymes and the transporter are well conserved between the clusters. The gene encoding LtnJ, however, responsible for the reduction in the conversion to alanine and butyric acid, was not conserved. Instead, a homolog to *sprOR* was found, suggesting it may carry out a similar function.
(PNG)

**S1 Table. *Streptomyces* genomes analyzed with decRiPPter.**
(XLSX)

**S2 Table. Precursor sequences of selected candidate RiPP families.** Serine and threonine residues are marked in green, and cysteine residues are marked in red.
(XLSX)

**S3 Table. Proteins containing a flavoprotein domain (PF02441) are present in both RiPP and non-RiPP BGCs.** While proteins with this domain are known in RiPP biosynthesis for the decarboxylation of carboxyl-terminal cysteines, their presence is not restricted to RiPP BGCs.
(XLSX)

**S4 Table. Homologs of the genes *lanJ$_A$*, *sprF1*, *sprF2*, and *sprOR* are found associated with both known lanthipeptide BGCs and close to the *sprPT/sprH3* gene pair.** Homology was determined at a cutoff of 30% amino acid identity of the gene products. Within *Streptomyces* genomes, all homologs were found within the analyzed 1,295 genomes. It was then checked whether these homologs overlapped with an antiSMASH-detected lanthipeptide BGC or were within 15 genes of the *sprPT/sprH3* gene pair. *sprOR* homologs were found within canonical lanthipeptide BGCs as well as associated with the *spriPT/sprH3* gene pair, suggesting its association with lanthipeptide BGCs. For non-*Streptomyces* genomes, the *sprPT/sprH3* gene pair was first detected, and homologs of the given queries were found within the 15 surrounding genes. Homologs of *lanJ$_A$* and *sprF1* are often found associated with *sprPT/sprH3* gene pair, suggesting they are involved in lanthipeptide biosynthesis.
(XLSX)

**S1 Data. Description of SVM training data and comparison with antiSMASH and RODEO.** (A) RiPP classes in positive training data of decRiPPter. (B) decRiPPter detects most RiPP precursors of known classes found by RODEO. RODEO results were extracted from previous studies [26,27,29–31]. (C) Comparison of BGCs mined by RODEO to decRiPPter detected. Note that not all genomes were analyzed by RODEO. Results from earlier RODEO genome mining [26,27,29–31] were only used if within the 1,295 *Streptomyces* genomes. (D) Comparison of antiSMASH-detected BGCs with decRiPPter-detected BGCs per RiPP class.
(XLSX)

**S2 Data. Categorized Pfam and TIGRFAM domains used in decRiPPter pipeline.**
(XLSX)

**S3 Data. Numerical data of all plotted graphs.**
(XLSX)

**S4 Data. Numerical data supporting the structural elucidation of pristinin A3 (1).** (A) Peaks unique to the extracts of strains containing pAK1 appear to be mostly derived from a single mass. (B) Many detected masses from strains containing the expression construct pAK2 appear to be derived from 2 masses. The 2 base masses were also the most abundant, making it likely these form final products, while the other masses may be incompletely processed products. (C) Observed masses for fragments of a mass of 2,703.235 Da can be matched to the SprA3 precursors. See also S11A Fig. (D) Observed masses for fragments of a peak corresponding to a monoisotopic mass of 2,553.260 Da can be matched to the SprA2 precursor. See also S11B Fig. (E) $^{1}$H and $^{13}$C NMR data for pristinin A3 (DMSO-$d_6$, 850 MHz, 298 K, and $^{1}$H and $^{13}$C NMR data for the G24, A25, and the carboxyl-terminal ring of pristinin A3 (CD$_3$CN:H$_2$O 9:1, 850 MHz, 297 K)). See also S13, S14 and S15 Figs. (F) Ratio of oxidized product in samples analyzed by NMR. (G) Fragmentation data of oxidized products. A mixture of oxidized and nonoxidized fragments can be observed when the fragments do not contain the center ring structure. When the fragments do contain the center ring structure, they are always oxidized, suggesting the center ring contains the oxidation. (H) Cysteines linked to serine and threonine residues are detected after acidic hydrolysis of SprA3. Below are the predicted amino acids and their detected masses. Most of the amino acids can be detected in the chromatogram,

including the cysteines linked to dehydrated serine and threonine residues. The mass of the predicted decarboxylated cysteine linked to a dehydrated threonine residue was not detected, nor were the dehydrated serine and threonine residue. However, given that these groups contain alkenes, which easily react under acidic conditions, these groups may have been degraded. (XLSX)

**S5 Data. Primers and plasmids used in this study.**
(XLSX)

**S1 Text. Alignment of precursors belonging to the characterized family of type V lanthipeptides.** Precursors were aligned with MUSCLE [101] and visualized with BoxShade. (DOCX)

## Author Contributions

**Conceptualization:** Peter Cimermancic, Mohamed S. Donia, Michael A. Fischbach, Gilles P. van Wezel, Marnix H. Medema.

**Data curation:** Alexander M. Kloosterman, Peter Cimermancic, Marnix H. Medema.

**Formal analysis:** Alexander M. Kloosterman, Peter Cimermancic, Somayah S. Elsayed, Chao Du, Michalis Hadjithomas, Michael A. Fischbach, Gilles P. van Wezel, Marnix H. Medema.

**Funding acquisition:** Gilles P. van Wezel.

**Investigation:** Alexander M. Kloosterman, Peter Cimermancic, Somayah S. Elsayed, Chao Du, Michalis Hadjithomas, Mohamed S. Donia, Michael A. Fischbach, Marnix H. Medema.

**Methodology:** Alexander M. Kloosterman, Peter Cimermancic, Somayah S. Elsayed, Michalis Hadjithomas, Mohamed S. Donia, Michael A. Fischbach, Marnix H. Medema.

**Project administration:** Gilles P. van Wezel.

**Resources:** Alexander M. Kloosterman, Peter Cimermancic.

**Software:** Alexander M. Kloosterman, Peter Cimermancic, Michalis Hadjithomas, Marnix H. Medema.

**Supervision:** Michael A. Fischbach, Gilles P. van Wezel, Marnix H. Medema.

**Validation:** Alexander M. Kloosterman, Peter Cimermancic, Marnix H. Medema.

**Visualization:** Alexander M. Kloosterman.

**Writing – original draft:** Alexander M. Kloosterman, Gilles P. van Wezel, Marnix H. Medema.

**Writing – review & editing:** Alexander M. Kloosterman, Peter Cimermancic, Gilles P. van Wezel, Marnix H. Medema.

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
