## [Editor Report · Decision Letter 0]

29 May 2020

Dear Dr Medema, 

Thank you for submitting your manuscript entitled "Integration of machine learning and pan-genomics expands the biosynthetic landscape of RiPP natural products" for consideration as a Methods and Resources by PLOS Biology.

Your manuscript has now been evaluated by the PLOS Biology editorial staff, as well as by an academic editor with relevant expertise, and I'm writing to let you know that we would like to send your submission out for external peer review.

Please re-submit your manuscript within two working days, i.e. by Jun 02 2020 11:59PM.

Kind regards,

Roli Roberts

Senior Editor

PLOS Biology

---

## [Decision Letter · Decision Letter 1]

20 Jul 2020

Dear Dr Medema,

Thank you very much for submitting your manuscript "Integration of machine learning and pan-genomics expands the biosynthetic landscape of RiPP natural products" for consideration as a Methods and Resources paper at PLOS Biology. Your manuscript has been evaluated by the PLOS Biology editors, an Academic Editor with relevant expertise, and by four independent reviewers.

You'll see that all four reviewers are broadly positive about your study, but three of them raise a number of issues that should be addressed. In particular, rev #4 has some requests that involve new experimental data and some analyses; this reviewer and rev #2 have some additional extensive textual requests; all concerns should be addressed for further consideration.

In light of the reviews (below), we will not be able to accept the current version of the manuscript, but we would welcome re-submission of a much-revised version that takes into account the reviewers' comments. We cannot make any decision about publication until we have seen the revised manuscript and your response to the reviewers' comments. Your revised manuscript is also likely to be sent for further evaluation by the reviewers.

We expect to receive your revised manuscript within 2 months. 

**IMPORTANT - SUBMITTING YOUR REVISION**

*Re-submission Checklist*

*Published Peer Review*

*PLOS Data Policy*

*Blot and Gel Data Policy*

Sincerely,

Roli Roberts

Senior Editor

PLOS Biology

REVIEWERS' COMMENTS:

Reviewer #1:

In this manuscript, the authors present a very well-crafted and particularly useful bioinformatics tool to identify a class of biosynthetic genes that - so far - proved difficult to identify in bacterial genomes. This tool, DecRiPPter, allows to perform genome mining in order to pinpoint ribosomally synthesized and post-translationally modified peptides (RiPPs). Whereas biosynthetic genes that are associated with e.g. polyketides or nonribosomal peptides can be identified using class-specific markers of highly conserved domains, this is virtually impossible for RiPPs, which lack these necessary markers. The authors tackle this difficult problem with a powerful machine learning and a pan-genomics approach rather than relying on known tailoring enzymes. This has the advantage that novel RiPPs can be found rather than 're-discovering' RiPPs that are merely modifications of known ones. This approach is followed by a second layer of core and accessory profiling at the BGC level and then a network analysis with antiSMASH results. 

The bioinformatics approach is reasonable and it is tested on isolate genomes not only to find novel RiPPs but they also chemically characterize them to show that they belong to a novel class thus confirming its proposed utility.

Overall, this is a very elegant approach whose applicability and usefulness has been nicely demonstrated. 

Some minor points: 

1) The authors focused on bacteria of the genus streptomyces yet natural products researchers would certainly like to use this tool for other genera such as Pseudomonas. It would be highly appreciated if the authors could comment if this tool also works for these bacteria. 

2) The list of references is missing some bibliographical information. In particular references 21, 22, 25, 39, 54, 58, 60, 61, 71, 75 (e.g. journal names missing, etc.). Please update the bioRxiv citations 29, 30, and 76 which are all already published. Reference 16 and 34 is identical. Please re-check all references.

3) Lines 163-167: could be rephrased for better readability

4) Lines 344, 345, 385: please italicize 'S', 'Z', and 'm/z'

5) Line 378: please change 'likely cryptic' to 'likely silent'

6) Line 470: remove '.'

7) Line 362: The SI mentioned pAK3. Please include the plasmid name in manuscript.

8) Line 394-396: please include corresponding method description in the SI

9) There is a recent paper in Angew Chem (https://doi.org/10.1002/anie.202005187) that describes a novel RiPP - Cacaoidin. This compound seems to bear some resemblance to the RiPP described in this manuscript. Of course this does not influence the novelty of the authors' work yet it should be mentioned/addressed in the manuscript.

Reviewer #2:

In this study, the authors focused on developing new bioinformatic methodology to find RiPP biosynthetic gene clusters (BGCs) that are not readily found by traditional means of looking for novel RiPP BGCs which mostly rely on knowledge of the PTM enzymes. The method is interesting as it could potentially discover new RiPP classes. 

The method looks interesting to me, using a stepwise approach to finding such novel BGCs. The authors acknowledge that it could lead to many false positives and I agree, but if used as it is here, by growing up a potential candidate and looking for new compounds, I believe it could be valuable to prioritize candidates. The authors also acknowledge that true RiPP BGCs are also likely filtered out in their workflow, but at this stage I think that is OK. I must say that I am not a specialist in bioinformatic tool development. Hence my comments are those of a potential would-be user.

The authors mention that the last step is dereplication compared to known RiPP BGCs. They mention they used MIBiG and antiSMASH to do so. It must be said that neither of these are comprehensive at present and that classes may well be missed (as also indicated by lines 262-33). Can the authors state which currently known RiPP classes are being dereplicated by the current strategy? Are the currently known ~40 RiPP classes all represented in MIBiG or are representative examples for all found by antiSMASH?

The authors mention in the introduction that their method can potentially find clusters not found when using known PTM enzymes as queries using existing methods. But it must be said that the compound they describe could probably have been found using either the known family of Cy decarboxylases or the combination of SprH3/PT (which had already been suggested as the likely dehydratases for thioviridamide and linaridin biosynthesis). Hence, while I recognize that there are still many other candidate gene cluster families that the authors have not characterized, I think it should be mentioned that previous methodology (e.g. RODEO and perhaps DEEPRiPP) could have worked to find these class V lanthipeptides when applying the correct query. I think the authors should at least comment on this. I don't think that diminishes the value of this study, but it would place it better in context of other studies. Another prior approach that deserves mentioning in the introduction is looking for precursor genes near genes for AMC transporters (PLoS One. 2014;9(3):e91352. doi: 10.1371/journal.pone.0091352.). Although that study focused on lanthipeptides, and the current work obviously is much more comprehensive, I think it is a strategy that can be run agnostic of RiPP class. 

Using their new platform, the authors discovered a new lanthipeptide and link it to a BGC using genetic studies that are well executed and quite convincing. Because the BGC does not contain the typical class I-IV lanthipeptide biosynthetic genes, the authors suggest this is a new class (class V to follow previous nomenclature first introduced by Sahl, although the current authors use "type" V in this paper; I recommend to be consistent and call them class V). This is exciting and interesting. The authors do not comment on a very recent paper that came to the same conclusions based on an activity-based fractionation approach that resulted in a new lanthipeptide (although the authors of that paper did not call the new group class V, it is clear its BGC has all the hallmarks of the current BGC and the structure of the two compounds are related). While I recognize that the current work was ongoing when that paper appeared, I still think the authors should discuss this other paper and refer to it (Angew Chem Int Ed Engl. 2020 doi: 10.1002/anie.202005187.). 

The suggestion that SprOR-like proteins may be responsible for formation of D-amino acids was also reached in another recent paper on lanthipeptide genome mining by isolating a two-component lanthipeptide with Ser-to-Ala conversions. Its BGC lacks LanJ but encodes an SprOR-like protein in its BGC, thus providing direct support for the hypothesis in Fig S18: BMC Genomics. 2020 21(1):387. doi: 10.1186/s12864-020-06785-7.

Overall, I think this study is well suited for PLoS Biol after the authors address the comments above as well as specific comments below.

Specific comments:

Abstract "Most clinical drugs are based on microbial natural products". This statement is too general. It is true in some disease areas (e.g. infectious diseases) but is not correct for all clinical drugs.

Are fluoroquinones (perhaps better fluoroquinolones?) really natural product derived? I think they are purely synthetic and not inspired or derived from natural products (their history goes back to an impurity in synthetic efforts to chloroquine synthesis).

P3 "homologues of RiPP tailoring enzymes (RTEs) of interest". This nomenclature is not following the recommended nomenclature for RiPPs (ref 19). In the PKS/NRPS world the term tailoring is used for reactions that are compound specific and not class determining (they usually take place after the class determining reactions). In Arnison et al in 2013, the same use was recommended: tailoring reactions in RiPP biosynthesis are not class defining, but specific for a certain compound. Hence the definition here for RTEs is awkward as it goes against the community-agreed nomenclature. Class-defining or primary-modifications is what researchers in this area have used. I think it would be counterproductive to keep the nomenclature as it is introduced here (RTEs). Perhaps RPEs (RiPP primary-modification enzymes)?

P3, line 109 "by a previously unknown biosynthetic machinery". The machinery has been known in linaridin and thioviridamide biosynthesis (and in those cases proposed to be involved in dehydration). So, while it was not previously known for lanthipeptides, the machinery itself was certainly known (albeit not characterized, but it is also not characterized in this work).

P4 "To make sure that this classifier could predict precursors independent of RiPP subclass, we trained it on all possible subsets of the positive training set in which one of the RiPP subclasses was entirely left out". It would be helpful to show/discuss which subclasses the authors are referring to here. Indeed, in 2013 ~ 20 classes of RiPPs were known as the authors mention but that number is almost doubled (perhaps more than doubled?) and the current wording does not make clear which classes were used.

Line 140. Define COG acronym at first use (now defined in line 172)

Line 141. "are then analyzed" would be better as "were then analyzed" (the remainder of the paragraph is all in past tense).

Line 175. Define BUSCO at first use.

"Families in which more than half of the gene clusters overlapped with antiSMASH non-RiPP BGCs were discarded as well," The authors might consider including a sentence that this step may eliminate BGCs that produce hybrids of RiPP and non-RiPP pathways which have been reported (e.g. ref 23 and Nat Chem Biol. 2018 Jul;14(7):652-654. doi: 10.1038/s41589-018-0068-6.)

Line 308. Replace "sulfide" with "thiol" (sulfide is S2-)

Line 386. "showed that the six of the corresponding compounds"

Line 451: For consistency perhaps replace lantibiotics (only used here) with lanthipeptides (used throughout the paper).

Line 454: LanKL is not correct. This should be LanL.

Line 470. "available. , strongly supporting"

Line 482: "which is often associated with small prepeptides such as methanobactins". Methanobactins are not precursor peptides. They are the final RiPP.

The referencing is incomplete or outdated in many instances. This must be fixed prior to publication.

Many references are missing page numbers.

Ref 29. This bioRxiv paper from 2018 has been published since. Please replace. Nucleic Acids Res. 2019 May 21;47(9):4624-4637. doi: 10.1093/nar/gkz192

Ref 30. This bioRxiv paper from 2019 has been published since. Please replace. Sci Rep . 2019 Sep 16;9(1):13406. doi: 10.1038/s41598-019-49764-z. 

Ref 39, 60, 61, 71, 75 is incomplete (missing year, volume, page numbers)

Ref 53: Referencing is incomplete. there are three other groups that reported on bottromycin in the same year in addition to this work (some of the others submitted earlier if the authors are to select just one; I think they should all be referenced).

Figure 2. phsophatase

Table 1. sprPT1 is suggested to be involved in Cys decarboxylation (last column). Why? LanD enzymes (SprF) do this without needing a phosphotransferase. 

Reviewer #3:

The authors describe the development of an algorithm and the program decRiPPter that allows predicting the biosynthetic gene clusters of ribosomally synthesized and post-translationally modified peptides (RiPPs). In contrast to existing methods, their new algorithm was specifically designed to detect also novel RiPP classes. They applied this algorithm to Streptomyces genomes and detected 42 new candidate RiPPs. One of them in S. pristinaespiralis was characterized in more detail and resulted in the discovery the first member of a new family of lanthipeptides. The distribution of two core genes of this family, the sprPT/sprH3 gene pair was found in many BGCs with more or less homology of the general BGC architecture. The elucidation of a novel RiPP is definitely a great discovery, yet, the merit of this manuscript and the underlying work is to been seen more in the development of decRiPPter which has been made freely available and the discovery that there are many more RiPPs than could have been previously identified by antiSMASH or other programs. The only point I would like to rise is that it would have been preferable if the program decRiPPter could have been made available as web-based tool just like antiSMASH which would increase the user friendliness and the acceptance.

This may be an important tool for many researchers interested in natural products and give rise to the discovery of many novel RiPPs and RiPP-families. 

The manuscript is well written and the research of highest quality. I have thus no doubts, that this manuscript should be accepted for publication in PLOS Biology. 

Reviewer #4:

This manuscript represents a very ambitious effort in identifying new RiPP biosynthetic gene clusters. Here, the authors have developed a series of bioinformatic rules (associated with a new tool, decRiPPter) that potentially represent a real advance in how gene clusters are identified. The analysis of conserved orthologous genes (COGs) seems a really neat way of identifying possible secondary metabolite clusters (RiPPs or otherwise), and the rules the authors define to identify likely RiPP gene clusters seem logical for the majority of clusters and does yield some really interesting candidate RiPP clusters. Below I will highlight both positive and negative aspects of the manuscript. On balance, I believe this will necessitate significant revisions to the manuscript prior to publication.

1. There are some real innovations in how it identifies new clusters, and the use of publicly available datasets to validate and refine the cluster mining approach is done very well. The ambition here is to be praised too, given that the focus is on novel clusters rather than new versions of existing RiPP families. 

2. The online data is really appreciated and is well presented. This provides a substantial dataset for other researchers to investigate. It would benefit from a few improvements and corrections. Firstly, it would be much easier to navigate if strain names were associated with entries and/or there was a connection in some way to the data presented in the manuscript. For example, it wasn't easy to connect the 42 RiPP families described in the paper to the online information. One apparent minor error is that the genes in the GenBank files provided online seem to be in the wrong frame (forward strand only) when opened in gene viewer software. Also, the filter function doesn't seem to work online.

3. A major concern is the likelihood of false positives that arise. The manuscript partially addresses this, but should do more to highlight and analyze this possibility for any potential users. An analysis of the online "strict" mode dataset picks out a number of clusters that are unlikely to be RiPPs:

- Clusters containing several consecutive phage structural proteins: 4 different ones, associated to 6 different precursor groups (1211, 22163, 4804, 2244, 1821, 6306)

- Clusters including a high proportion of sugar metabolism protein domains: precursor groups 5606, 10155, 76415, 12804, 50706, 170, 11811, 7713, 73777, 84405, 19730, 64989, 83901, 27291, 3156, 27600, 18839, 62573, 74379 (and maybe some more). Some of them could be real RiPP clusters, but it is possible that many of them are not

- Clusters that look like textbook RiPPs, such as RBGC_497026 (belonging to the large precursor group 462), but further domain analysis shows that the precursor has a LysW domain, which is characteristic of a carrier protein involved in lysine biosynthesis (http://www.nature.com/articles/nchembio.198). Other proteins encoded in the precursor group 462 clusters have similarity to this lysine biosynthesis pathway.

It is possible that some of these are filtered in the analysis described in the manuscript but not in the online data, as line 267 mentions that LysW peptides were discarded, yet do appear as "true" precursors via the strict mode online. Therefore, either some inconsistencies in the online data need correcting, or the authors should attempt to devise ways to further validate their findings and/or discuss false positive likelihood in the text. Conversely, are there clusters that were picked up in previous genome mining studies that are not identified by decRiPPter but might have been expected?

4. Related to point 3, some further validation of the precursor peptide identification should be carried out. Due to the diversity of precursor peptide sequences, it seems likely that there will be a mixture of false positives (e.g. the examples mentioned in point 3) as well as missed precursor peptides when using an SVM approach. This isn't necessarily a big problem as the study aims to highlight unusual clusters, so some mis-called precursors seem acceptable, but are there any further approaches the authors could take to try to determine whether a predicted precursor peptide gene is likely to be real or not? For example, how many are already annotated as genes? If not, why not? A comparison with other precursor peptide assessment tools seems relevant here (e.g. NeuRiPP and DeepRiPP).

5. The authors make a significant claim about the discovery of type V lanthipeptides, but this needs further experimental validation given that the molecule has not been properly characterized. The manuscript has a very lengthy section on the attempts to identify the product and then partially characterize it, but simply needs proper NMR analysis of the product. The proper chemical characterization of one or more molecules in a mining paper like this is consistent with comparable studies (e.g. PNAS, 2020, 117, 371; NCB, 2017, 13, 470). This is especially important as a new class is being proposed, which includes speculation on a new set of proteins involved in the formation of D-amino acids. 

6. In relation to the designation of type V lanthipeptides, the gene pair that are described (sprH3/sprPT) are also present in clusters for the thioviridamides, which are not lanthipeptides. A qualification of this "type V" designation is therefore necessary - is it possible that some molecules are cyclized like lanthipeptides while others are dehydrated (or epimerized, based on the hypothesis associated with SprOR)? Or do these genes do something completely different given the lack of experimental data?

7. This may reflect a lack of experience with the interface, but it was difficult to find the spr gene cluster via the online interface. A search for the Streptomyces pristinaespiralis genome provided this output, which doesn't seem to include the spr cluster: http://www.bioinformatics.nl/~medem005/decRiPPter_strict/Subgroups/genome/operons_genome_GCF_001278075.1.html

8. The data presented for the characterization of the spr product could be improved in the main manuscript. For example, Figure 3C shows a total ion current but doesn't directly relate this to the masses described in the text, while the reader has to go deeply into the SI to see full chromatograms of the various mutants (regulator over-expressed, delta-spr etc). The volcano plot presented in Figure S13 looks impressive, but it is difficult to understand what is presented - which peaks represent the predicted RiPP, what are the other peaks, and what does gray and pink denote? Improved presentation of this data would help guide the reader to the conclusions about the RiPP product. 

Further minor comments/corrections:

Line 14: I believe fluoroquinones were not inspired by natural products and represent a truly synthetic class of antibiotic.

Lines 167-172: the explanation here could be rephrased and simplified for non-experts. Would a figure help?

Line 193: "Namely" should be substituted by "For example", as some RiPP classes produced by Streptomyces are not included in that list (e.g. pheganomycin).

Lines 338-339: the differences between the spr precursors could be better visualized (currently embedded within a much larger alignment in Figure S10).

Line 377-378. the manuscript reads as if only one medium was tested for production. If so, the statement that the spr cluster is cryptic should be removed. Is there any evidence that the cluster is not expressed under many different conditions? If only one medium is tested, many gene clusters are likely to be "cryptic".

Line 498: some more context for LtnJ and CrnJ would be useful to guide the reader - what is their conserved domain and what pathways are they from?

Line 504: "thus candidates" corrected to "thus is a candidate"

Line 506: "Table S10" corrected to "Table S11"

Table 2 needs correcting in places, such as naming of SprPT1 (SprF1?) and SprPT2, and the domain of SprF1.

---

## [Decision Letter · Decision Letter 2]

11 Nov 2020

Dear Dr Medema,

Thank you for submitting your revised Methods and Resources paper entitled "Integration of machine learning and pan-genomics expands the biosynthetic landscape of RiPP natural products" for publication in PLOS Biology. I have now obtained advice from one of the original reviewers and have discussed their comments with the Academic Editor. 

Based on the reviews, we will probably accept this manuscript for publication, assuming that you will modify the manuscript to address the remaining points raised by the reviewer. Please also make sure to address the data and other policy-related requests noted at the end of this email.

IMPORTANT:

a) Please attend to the remaining requests from reviewer #4.

b) Regarding this reviewer's comments about supplementary files, on the one hand, we do like the ability to link to specific components of the supplement, but on the other hand we strongly agree with the reviewer's comment that these could be more helpfully organised. Specifically, we suggest:

 1. Some of the Supplementary Figures are small and thematically linked. These should be combined into a single multi-panel Figure. This is recommended.

 2. Many of the Supplementary Tables are very small, very simple in structure, and thematically linked. These should be combined on single spreadsheets or on multiple sheets in single files. This is very strongly recommended.

 3. Supplementary Protocols 1 and 2 (and probably 3) should be incorporated into the main manuscript as part of a formal Materials and Methods section. This is essential.

c) Please attend to my Data Policy requests further down this email.

d) Please could you change the title to something more explicit, such as "decRIPPter: integration of machine learning and pan-genomics expands the biosynthetic landscape of the RiPP natural product class"?

We expect to receive your revised manuscript within two weeks. Your revisions should address the specific points made by each reviewer. In addition to the remaining revisions and before we will be able to formally accept your manuscript and consider it "in press", we also need to ensure that your article conforms to our guidelines. A member of our team will be in touch shortly with a set of requests. As we can't proceed until these requirements are met, your swift response will help prevent delays to publication.

- a cover letter that should detail your responses to any editorial requests, if applicable

*Copyediting*

*Published Peer Review History*

*Early Version*

Sincerely,

Roli Roberts

Senior Editor,

rroberts@plos.org,

PLOS Biology

DATA POLICY:

Regardless of the method selected, please ensure that you provide the individual numerical values that underlie the summary data displayed in the following figure panels as they are essential for readers to assess your analysis and to reproduce it: Figs 3BCDE, 4, S3, S4, S5, S6, S7, S9, S10ABCDEF, S11ABC, S12, S13, S14ABC, S15, S16, S18-S33. NMR data should be deposited appropriately. NOTE: the numerical data provided should include all replicates AND the way in which the plotted mean and errors were derived (it should not present only the mean/average values).

REVIEWER'S COMMENT:

Reviewer #4:

The majority of referee comments have been thoroughly answered and suitably addressed within the revised manuscript. This represents a tool that could be really valuable, as well as the characterization of an interesting new RiPP natural product. I believe the manuscript is very suitable for publication now, where there are simply a few minor typographical errors that need addressing (Line numbers relate to the "R2" PDF that was supplied). In my opinion, these edits do not need further review:

* Line 220: Space needed between sentences

* This may be the journal style, but "S3 Figure A" (Line 227) and similar supporting information nomenclature sounds odd

* Line 276: there appears to be a missing number in "7.18 *10" 

* Line 424: Cysteine should be lower case spelling

* Figure S12: The text for each row is almost unreadable. Advise making the rows larger or text slightly smaller

* Line 525: There is some detail that seems unnecessary, such as "Even though the NMR shimming was appropriate, as evidenced from the solvent peak". The authors could simply say that NH peaks were not visible in DMSO, so CD3CN:H2O was used to obtain clear NH signals

* It seems appropriate to name compound 1.

* Line 696: Re-word: "tools aimed at novelty may discovery new natural products". Perhaps: "tools aimed at novelty may lead to the discovery of new natural products".

There are also a couple of additional minor comments:

1. It is understandable that full stereochemical assignments aren't provided for compound 1, but cacaoidin includes D-alanine so is it worth commenting on this possibility for compound 1? It looks like there are serine residues that are converted to alanine residues, which would enable this.

2. I'm unsure whether this is a journal decision or an author decision, but the organization of data for this re-submission made reviewing much more difficult than should be necessary. If data is presented like this for the published paper, it is likely that a reader will ignore a lot of great supporting data. The following files had to be downloaded and viewed individually (I thought I'd list them all to show how much this is). I think this is all relevant data that deserves to be associated with the publication, but a single supporting info file would be much easier to follow (text, protocols, tables and embedded figures alongside their figure legends), as would a single Excel spreadsheet with multiple sheets (or a limited number of Excel files with multiple sheets).

Supporting files:

S1_Text

S1_Data

S1_Figure

S2_Figure

S3_Figure

S4_Figure

S5_Figure

S6_Figure

S7_Figure

S8_Figure

S9_Figure

S10_Figure

S11_Figure

S12_Figure

S13_Figure

S14_Figure

S15_Figure

S16_Figure

S17_Figure

S18_Figure

S19_Figure

S20_Figure

S21_Figure

S22_Figure

S23_Figure

S24_Figure

S25_Figure

S26_Figure

S27_Figure

S28_Figure

S29_Figure

S30_Figure

S31_Figure

S31_Figure

S32_Figure

S33_Figure

S34_Figure

S35_Figure

S1_Table

S2_Table

S3_Table

S4_Table

S5_Table

S6_Table

S7_Table

S8_Table

S9_Table

S10_Table

S11_Table

S12_Table

S13_Table

S14_Table

S15_Table

S16_Table

S17_Table

S18_Table

S19_Table

S1_Protocol

S2_Protocol

S3_Protocol

---

## [Editor Report · Decision Letter 3]

7 Dec 2020

Dear Dr Medema,

On behalf of my colleagues and the Academic Editor, Tobias Bollenbach, I am pleased to inform you that we will be delighted to publish your Methods and Resources in PLOS Biology. 

PRODUCTION PROCESS

Before publication you will see the copyedited word document (within 5 business days) and a PDF proof shortly after that. The copyeditor will be in touch shortly before sending you the copyedited Word document. We will make some revisions at copyediting stage to conform to our general style, and for clarification. When you receive this version you should check and revise it very carefully, including figures, tables, references, and supporting information, because corrections at the next stage (proofs) will be strictly limited to (1) errors in author names or affiliations, (2) errors of scientific fact that would cause misunderstandings to readers, and (3) printer's (introduced) errors. Please return the copyedited file within 2 business days in order to ensure timely delivery of the PDF proof. 

If you are likely to be away when either this document or the proof is sent, please ensure we have contact information of a second person, as we will need you to respond quickly at each point. Given the disruptions resulting from the ongoing COVID-19 pandemic, there may be delays in the production process. We apologise in advance for any inconvenience caused and will do our best to minimize impact as far as possible.

EARLY VERSION

PRESS 

Kind regards,

Alice Musson

Publishing Editor, 

PLOS Biology

on behalf of

Roland Roberts,

Senior Editor

PLOS Biology